# $\mathcal{B}$-CODER: VALUE-BASED DEEP REINFORCEMENT LEARNING FOR PROGRAM SYNTHESIS

**Zishun Yu**[*]
Department of Computer Science
University of Illinois Chicago
Chicago, IL 60607
zyu32@uic.edu

**Yunzhe Tao, Liyu Chen, Tao Sun & Hongxia Yang**
ByteDance Inc.
Seattle, WA 98004
{yunzhe.tao, liyu.chen1,
tao.sun, hx.yang}@bytedance.com

## ABSTRACT

Program synthesis aims to create accurate, executable programs from problem specifications, specifically from natural language descriptions in our context. Recent studies have leveraged the power of reinforcement learning (RL) in conjunction with large language models (LLMs), significantly enhancing code generation capabilities. The application of RL focuses on directly optimizing for functional correctness, offering an advantage over conventional supervised methods. Despite policy-based RL methods dominating the literature on RL for program synthesis, the nature of program synthesis tasks hints at a natural alignment with value-based methods. This stems from the rich collection of off-policy programs, including those developed by human programmers and also historical samples, coupled with the straightforward verification of generated programs through automated unit testing, meaning rewards are easy to obtain. Diverging from the dominant use of policy-based algorithms, our work explores the feasibility of value-based approaches, leading to the development of our $\mathcal{B}$-Coder (pronounced Bellman coder). Yet, training value-based methods presents challenges due to the enormous search space inherent to program synthesis. To this end, we introduce an initialization protocol for RL agents utilizing pre-trained LMs and a conservative Bellman operator to reduce training complexities. Moreover, we demonstrate how to leverage the learned value functions as a dual strategy to post-process generated programs. Our empirical evaluations demonstrated $\mathcal{B}$-Coder's capability in achieving state-of-the-art performance when compared to policy-based methods. Remarkably, this achievement is reached with minimal reward engineering effort, highlighting the effectiveness of value-based RL, independent of reward designs.

## 1 INTRODUCTION

Program synthesis (or code generation) aims to create functionally accurate executable programs from problem specifications, such as input-output (IO) examples (Summers, 1977; Gulwani et al., 2012), constraint-based (Osera & Zdancewic, 2015; Frankle et al., 2016) or natural language descriptions (Hendrycks et al., 2021; Austin et al., 2021), among others. The increasing attention towards this field can be attributed to its potential in transforming the software development paradigm. Notably, AI-powered tools have shown evidence of boosting efficiency within the software industry.

Large language models (LLMs) (Brown et al., 2020; OpenAI, 2023; Anil et al., 2023; Chowdhery et al., 2022; Rae et al., 2021; Hoffmann et al., 2022; Touvron et al., 2023) have garnered substantial interest and shown remarkable achievements. The scheme of pre-training on vast amounts of data has yielded notable successes in natural language generation. This trend extends its influence to program synthesis, where numerous specialized code LLMs (Li et al., 2023; 2022; Nijkamp et al., 2022; Zheng et al., 2023; Fried et al., 2022; Chen et al., 2021a; Wang et al., 2021; 2023; Xu et al., 2023; Rozière et al., 2023) have been introduced to address challenges in program synthesis.

Unlike many free-form natural language generation tasks, where the quality of model's output is hard to assess, the correctness of synthesized programs can be verified through automated execution with

---

[*]This work was done during an internship at ByteDance Inc. Correspondence to Zishun Yu.

predefined unit tests. This allows for directly optimizing execution outcomes through reinforcement learning (RL), by formulating test outcomes as reward signals. Our discussion focuses on recent RL-based works (Le et al., 2022; Shojaee et al., 2023; Liu et al., 2023) that have achieved remarkable advancements in Python text-to-code generation, evaluated on the challenging benchmarks sourced from Codeforces programming contests (Hendrycks et al., 2021; Li et al., 2022) Notably, these works predominantly favor *on-policy* policy-based algorithms.

While (on-policy) policy-based methods are favored in existing program synthesis works, they are known to be sample inefficient (Nachum et al., 2017; Gu et al., 2016) due to their inability to use *off-policy samples*. In contrast, value-based methods, using temporal difference learning, are known to be more sample-efficient (Gu et al., 2016; Nachum et al., 2017; Liu et al., 2020), as they solve a fixed-point iteration which does not explicitly require a specific data distribution, hence offering better compatibility with off-policy data. We defer the technical explanations on on/off-policy data and reasons for the different efficiency to Section 3.2, where we have notations and definitions ready.

In program synthesis, the primary sources of off-policy data include human programs and previously synthesized programs. Both are off-policy as they do not follow the sequence distribution induced by the *current* model. Current program synthesis works often directly use off-policy samples with on-policy methods. Unsurprisingly, Shojaee et al. (2023) notices that an increase in off-policy synthetic programs may degrade performance. This occurs as off-policy data lead to biased gradient estimates. Ideally, an objective should be to enhance or at least sustain performance as data volume grows.

To summarize, the reasons that suggest a natural fit for value-based methods in program synthesis are twofold: the availability of (inexpensive) rewards, similar to classical RL tasks like GO and Atari; and the principle compatibility with off-policy data for effectively leveraging human and historical data. However, value-based RL faces challenges such as difficulty in converging in large state-action spaces. To this end, we introduce $\mathcal{B}$-Coder (Bellman coder), with our contributions being threefold:

- We stabilize value-based RL for program synthesis by proposing an initialization protocol for $Q$-functions and a conservative Bellman operator to mitigate the training complexities.
- We demonstrate how to leverage value functions as a dual strategy to improve generation.
- $\mathcal{B}$-Coder achieves strong empirical performance with minimal reward engineering, providing further insights of RL algorithm design independent of reward function designs.

**Paper structure.** We introduce related works and notations in Section 2 and 3. Section 4 details our method and the rationale behind our design choices. Specifically, Sections 4.1, 4.2, and 4.3 address the challenges of value function training by: leveraging task structure, providing effective $Q$-function initialization, and a conservative operator for stable yet less ambitious updates, respectively. Section 4.5 shows an additional benefit of value functions, and Section 5 shows our empirical results.

## 2 RELATED WORKS

**Execution-guided program synthesis.** The feasibility of verifying programs through test case outcomes has led to the line of execution-guided works (Chen et al., 2018; Zohar & Wolf, 2018; Chen et al., 2021b). While these efforts leverage execution feedback, they do not directly optimize towards higher execution success rate due to the inherent non-differentiability of execution outcomes.

**RL for general sequence modeling.** Supervised LM training, using next token predictions (NTP) or masked language modeling (Kenton & Toutanova, 2019), has recognized limitations. One prominent issue is the exposure bias: given that the training is done in a "teacher-forcing" manner (Bengio et al., 2015; Ranzato et al., 2015), errors tend to accumulate during testing due to auto-regressive generation. In contrast, prior works (Ranzato et al., 2015; Rennie et al., 2017) have demonstrated the efficacy of RL in addressing exposure bias and optimizing non-differentiable metrics, e.g. BLEU (Papineni et al., 2002) and ROUGE (Lin, 2004), by leveraging automatic scoring as reward function.

**RL for program synthesis.** Supervised losses also fall short when assessing the functional accuracy of synthesized programs (Hendrycks et al., 2021; Chen et al., 2021a). As such, relying solely on supervised learning for program synthesis is not ideal. As RL provides a pathway to directly optimize non-differentiable objectives, plentiful work (Zhong et al., 2017; Simmons-Edler et al., 2018; Ellis et al., 2019; Wang et al., 2022) have studied enhancing code generation through RL. For the works most related to ours: CodeRL (Le et al., 2022) adapted REINFORCE (Williams, 1992), a classic

policy gradient (PG) algorithm, along with the baseline trick for variance reduction and a supervise-trained reward model to alleviate the issue of sparse execution signals. In addition, they proposed a critic sampling strategy to refine and repair program based on the example unit tests feedback. PPOCoder (Shojaee et al., 2023) applied proximal policy gradient (Schulman et al., 2017, PPO) to fine-tune pre-trained LMs. In addition, they leverage the syntactic and semantic structure of code, such as syntax trees (Rabinovich et al., 2017) and data-flow graphs (Yasunaga & Liang, 2020), to improve reward function designs. RLTF (Liu et al., 2023) proposed an online training framework for program synthesis using policy gradient with heursitically-designed fine-grained rewards.

**Additional discussions.** Appendix D lists several RL applications, showing the analogies between program synthesis and tasks that benefit from value-based methods. In C, we extend the discussion on works that extend policy-based methods to an off-policy setting. Such attempts often involve training a value function, further highlighting our motivation for starting with value-based methods.

## 3 PRELIMINARIES

One could formulate the program synthesis task as a sequence-to-sequence generation task, where a model takes a problem description $D$ as input and outputs a program $\hat{W}$ which aims to achieve the functionality specified by $D$. A generated program $\hat{W} = (\hat{w}_0, \ldots, \hat{w}_T)$ is composed by a sequence of tokens $\hat{w}_t \in \mathcal{V}$. For brevity, we use *constant* $T$ to denote the sequence length although it could be a variable in practice, and $W$ to denote a program in general (both generated and ground truth). Let LM be an instance of LM, $\ell((w_{<t}, D), \cdot)$ be the logits layer (language modelling head) output, and $p(\cdot|w_{<t}, D)$ be the probabilistic distribution over the vocabulary $\mathcal{V}$ (computed by passing $\ell(\cdot, \cdot)$ through softmax), conditioned on a sequence $w_{<t}$ and $D$. Suppose $W^*$ is a ground truth program and $\mathcal{D}_{\text{train}}$ is the train set, conventionally LMs could be trained by minimizing the cross-entropy loss

$$\mathcal{L}_{\text{ce}}(p) = -\mathbb{E}_{W^* \sim \mathcal{D}_{\text{train}}} \log p(W^*|D) = -\mathbb{E}_{W^* \sim \mathcal{D}_{\text{train}}} \sum_t \log p(w_t^*|w_{<t}^*, D). \quad (1)$$

### 3.1 RL NOTATIONS

To make notations easier to interpret, we bridge program synthesis notations to standard RL ones. RL problems are typically formulated as Markov Decision Processes (MDPs) and an MDP $\mathcal{M}$ is often composed by a 5-tuple $\mathcal{M} = (\mathcal{S}, \mathcal{A}, \mathbb{P}, r, \gamma)$ which are state space, action space, transition function, reward function and discount factor, respectively. The discount factor $\gamma$ discounts future values to emphasize the near futures, and we use $\gamma = 0.999$ (which slightly prefers more concise solution). A (stochastic) transition function $\mathbb{P} : \mathcal{S} \times \mathcal{A} \to \Delta(\mathcal{S})$ is a distribution over $\mathcal{S}$ conditioned on a state-action pair $(s, a)$. In program synthesis, $\mathbb{P}$ is trivial as $s_{t+1} \equiv s_t \circ a_t$, where $\circ$ denotes concatenation.

**State and action.** In code generation context, an action $a_t$ is a token $\hat{w}_t$. Hence the action space $\mathcal{A}$ is the vocabulary $\mathcal{V}$. As the information used to generate token $\hat{w}_t$ is $(\hat{w}_{<t}, D)$, the state is hence defined as $s_t := (\hat{w}_{<t}, D)$. For a given $D$, the state space $\mathcal{S} = \mathcal{V}^T$. For brevity, we will mainly use $s_t, a_t$ rather than the $w_t$ notations, and sometimes omit the time index $t$ if it leads to no confusion. We will also use $s', a'$ to denote $s_{t+1}, a_{t+1}$ whenever only the relative temporal position matters.

**Policy.** A policy $\pi : \mathcal{S} \to \Delta(\mathcal{A})$ assigns an action distribution $\Delta(\mathcal{A})$ to any state $s \in \mathcal{S}$, meaning predicting a token $\hat{w}_t$ based on current sequence $\hat{w}_{<t}$ and the problem specification $D$. Prior works often define $\pi_\theta \equiv p_\theta$ and directly optimize LM parameters $\theta$ with PG methods. We however define $\pi := f(\theta, \square)$ to be a function of $\theta$ and other components $\square$, see details in Section 4.

**Reward function.** A reward function $r : \mathcal{S} \times \mathcal{A} \to \mathbb{R}$ determines reward of taking action $a_t$ at state $s_t$. We follow the reward design of Le et al. (2022) in equation 2. We may also use shorthand notation $r_t := r(s_t, a_t)$. Note that the reward is determined when the program $W$ is completed at $T$. Thus $r_t = 0$ if $t \neq T$ otherwise defined as equation 2.

$$r(W) = r(s_T, a_T) =$$
$$\begin{cases} +1.0, & \text{if } W \text{ passed all unit tests} \\ -0.3, & \text{if } W \text{ failed any unit test} \\ -0.6, & \text{if } W \text{ cannot be executed} \\ -1.0, & \text{if } W \text{ cannot be compiled} \end{cases}$$
$$(2)$$

**Value functions.** RL maximizes the discounted returns, $J(\pi) = \mathbb{E}[\sum_t \gamma^t r_t | \pi, \mathcal{M}]$. The state-action value function $Q^\pi : \mathcal{S} \times \mathcal{A} \to \mathbb{R}$ and the state value function $V^\pi : \mathcal{S} \to \mathbb{R}$, are defined recursively as:

$$V^\pi(s) := \mathbb{E}\left[\sum_{t=0}^\infty \gamma^t r_t | \pi, \mathcal{M}, S_0 = s\right] = \mathbb{E}_{a \sim \pi(\cdot|s), s' \sim \mathbb{P}(\cdot|s,a)}\left[r(s, a) + \gamma V^\pi(s')\right] \quad (3)$$

$$Q^\pi(s, a) := \mathbb{E}\left[\sum_{t=0}^\infty \gamma^t r_t | \pi, \mathcal{M}, S_0 = s, A_0 = a\right] = \mathbb{E}_{s' \sim \mathbb{P}(\cdot|s,a)}\left[r(s, a) + \gamma Q^\pi(s', \pi)\right], \quad (4)$$

where $Q(s, \pi) := \mathbb{E}_{a \sim \pi} Q(s, a)$. In addition, the advantage function is $A^\pi(s, a) := Q^\pi(s, a) - V^\pi(s)$.

## 3.2 VALUE-BASED RL AND DUELING DQN

Value-based algorithms especially the $Q$-learning family (Watkins & Dayan, 1992; Mnih et al., 2013; Van Hasselt et al., 2016; Bellemare et al., 2017) have achieved remarkable successes. A canonical framework of the $Q$-learning family iterates between policy evaluation and policy improvement:

$$\text{policy evaluation (PE):} \quad Q_k = \arg\min_Q \mathbb{E}_\mathcal{D}[Q_{k-1}(s, a) - (r + \gamma Q_{k-1}(s', \pi_{k-1}))]^2 \quad (5)$$
$$\text{policy improvement (PI):} \quad \pi_k = \arg\max_\pi Q_k(s, \pi(s)) \quad (6)$$

where $\mathcal{D}$ is an arbitrary dataset, the PE step estimates the previous policy $\pi_{k-1}$ using the Bellman equation (Bellman, 1966), and the PI step finds an improved $\pi_k$ by maximizing $Q_k$ estimates.

In particular, we build our framework on top of Dueling DQN (Wang et al., 2016, DDQN). In a nutshell, DDQN approximates $V(s)$ and $A(s, a)$ with separate heads, and run improvement and evaluation steps with $Q(s, a) = V(s) + A(s, a)$. This bifurcation enables a robust estimation of $V(s)$ without conflating with the actions, which subsequently ensures a stable learning of $A(s, a)$ given that it focuses solely on the relative values. As a consequence, DDQN often exhibits enhanced stability in training dynamics and improved generalization. In addition to the prior mentioned advantages, DDQN enables us to leverage a task structure that ground truth programs should attain highest advantages, therefore reducing the searching space, which we will elaborate on in Section 4.1.

**Remarks on sample efficiency.** We illustrate the inefficiency of policy-based methods using vanilla PG as an example. PG maximizes $J(\mu) := \mathbb{E}[\sum_t \gamma^t r_t | \pi_\mu, \mathcal{M}] \equiv \mathbb{E}_{W \sim \pi_\mu}[\sum_t \gamma^t r_t]$, with gradient $\nabla_\mu J(\mu)$ computed using the policy gradient theorem. This method requires training data $W$ drawn from the distribution induced by *current policy* $\pi_\mu$, hence called *on-policy*. Therefore, one should in principle generate new data and discard historical data *at every update*, leading to undesired sample inefficiency. In contrast, policy evaluation as in equation 5 works with arbitrary dataset $\mathcal{D}$.

## 4 ALGORITHMIC DESIGNS - ACCELERATING VALUE-BASED TRAINING

While value-based RL holds great promise, its training can be challenging due to the large action space $\mathcal{A} = \mathcal{V}$ and the high-dimensional state space $\mathcal{S} = \mathcal{V}^T$. This leads to a notably large $Q$-table of size $\mathcal{O}(|\mathcal{V}|^T)$. And the cardinality of policy space is $|\mathcal{A}|^{|\mathcal{S}|} = \mathcal{O}(|\mathcal{V}|^{|\mathcal{V}|^T})$, which grows doubly exponentially. Both challenges from large action spaces and high-dimensional state spaces are pivotal research topics in RL. The action space challenges are discussed by e.g. Dulac-Arnold et al. (2015); Tavakoli et al. (2018); Kalashnikov et al. (2018), while He et al. (2016); Nair et al. (2018), among others, considered the state spaces complexities. In particular, Silver (2015); Duan et al. (2016) commented on that the potentially better training stability of policy-based methods in these scenarios.

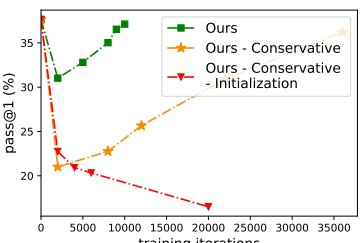

Figure 1: Training curves on APPS train set. ■ denotes $\mathcal{B}$-Coder, ⋆ removes our conservative operator, and ▾ is $\mathcal{B}$-Coder without both our operator and initialization.

To address the challenges inherent in training value-based RL for LMs, at a high level, we developed $\mathcal{B}$-Coder considering three key aspects: incorporation of task structure, initialization of $Q$-function, and backup using a conservative Bellman operator. Figure 1 previews the effectiveness of our algorithmic designs, which shows the training curve of different value-based RL algorithms on the APPS dataset. Due to aforementioned challenges, the performance of the vanilla DDQN continuously decreases even evaluated on the training set. In contrast, both the $Q$-function initialization and the conservative Bellman operator show benefits in stabilizing and accelerating the training process.

For notational convenience in subsequent sections, we begin with an overview of our notations and parameterizations, summarized in Figure 2. Figure 2(a) denotes a pre-trained encoder-decoder LM parameterized by $\theta_{\text{ckpt}}$ (where subscript ckpt denotes the fact it's a checkpoint/constant). Figure 2(b) and (c) show the forward graphs of our two different training stages: (b) corresponds to a pre-training stage for $\phi$, to provide a good initialization for (c) the subsequent fine-tuning of $\theta$. Motivations and details are deferred to Section 4.2 and 4.3, respectively. As we proceed to the rationale behind our

designs, it is encouraged to maintain familiarity with $\theta_{\text{ckpt}}$, $\phi$, $\theta$ and their corresponding products, especially the forward paths to $Q_\phi$ and $Q_\theta$, to prevent confusion in the subsequent sections.

## 4.1 LEVERAGING TASK STRUCTURES

As noted earlier, a key attribute of program synthesis task is the provision of human solutions, which are guaranteed to be correct. As a result, these solutions should attain the highest $Q$-values, even if the correct solutions might not be unique. As such, for a ground truth program $W^* = (s_0^*, a_0^*, \ldots, s_T^*, a_T^*)$, $Q(s_t^*, a_t^*) \geq Q(s_t^*, a)$ holds for all $a \in \mathcal{V}$, hence $A(s_t^*, a_t^*) \geq A(s_t^*, a)$.

To enforce this structure, one could ensure $A(W) \leq 0$ and $A(W^*) \approx 0$, where we abuse the notation and by letting $A(W) := \sum_{t=0}^T A(s_t, a_t)$. It ensures that $W^*$ has advantages that are roughly the highest. To this end, suppose $g(\cdot)$ is a general neural network, we decompose $Q$ as follows,

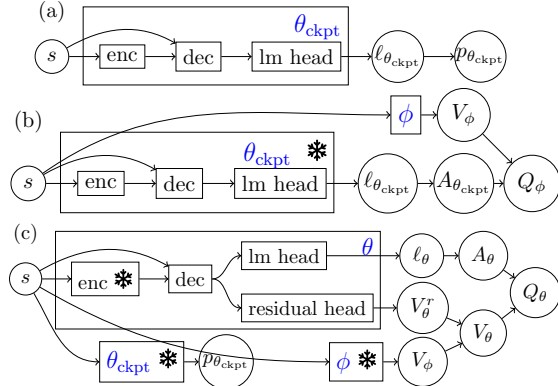

Figure 2: (a) A forward graph of conventional enc-dec LMs, with a checkpoint $\theta_{\text{ckpt}}$, where $p$ is a distribution over $\mathcal{A}$ and $\ell$ denotes logits ; (b) Our forward graph for pre-training $\phi$; (c) Our forward graph for fine-tuning $\theta$. ❄ indicates a frozen/constant component.

$$Q(s,a) = \underbrace{g(s,a) - \max_a g(s,a)}_{\text{non-positive advantage}} + V(s) = A(s,a) + V(s). \qquad (7)$$

It enforces our first condition that $A(W) \leq 0$. For the second condition $A(W^*) \approx 0$, we optimize an advantage function $A$ by minimizing an auxiliary advantage loss function, namely $\mathcal{L}_{\text{adv}}$,

$$\mathcal{L}_{\text{adv}}(A) = \mathbb{E}_{(s_0^*, a_0^*, \ldots, s_T^*, a_T^*) \sim \mathcal{D}_{\text{train}}} \left[ \sum_{t=0}^T |A(s_t^*, a_t^*)| \right]. \qquad (8)$$

We also cap the $Q$-function with $R_{\max} = 1$, the maximum total rewards. See Appendix G for details.

## 4.2 $Q$-FUNCTION INITIALIZATION

Despite the task structures introduced, training the $Q$-function *from scratch* remains extremely challenging. While this is not a problem for policy-based learning (given that directly fine-tune pre-trained LMs without requiring a $Q$-function at all), it presents significant challenges in value-based approaches because one often does not have a *pre-trained $Q$-function*. To this end, we show that one could initialize a $Q$-function from the logits output $\ell(\cdot, \cdot)$ of a pre-trained LM.

**Initialization of $Q$ via pre-trained models.** Yu & Zhang (2023) considered the fine-tuning of RL agents after offline RL pre-training. Their main idea is to reconstruct a $Q$-function from the pre-trained policy, for fine-tuning. Drawing inspiration from their approach, one could similarly reconstruct/initialize a $Q$-function using a pre-trained LM, akin to using a pre-trained policy.

This initialization was motivated by the energy-based policy line of works (Haarnoja et al., 2017; 2018), where a policy $\pi$ is the product of passing a $Q$-function through a softmax transfer function. Analogously, in LMs, $p$ - the distribution over $\mathcal{V}$ - is produced by passing logits $\ell$ through softmax.

$$\text{language modeling:} \quad p(a|s) = \exp(\ell(s,a)) / \sum_{a \in \mathcal{A}} \exp(\ell(s,a)) \qquad (9)$$

$$\text{energy-based } \pi: \quad \pi(a|s) = \exp\left(\tfrac{1}{\alpha}Q(s,a)\right) / \sum_{a \in \mathcal{A}} \exp\left(\tfrac{1}{\alpha}Q(s,a)\right), \qquad (10)$$

where $\alpha$ is a temperature hyper-parameter. One could naturally set $Q(s,a) = \alpha\ell(s,a)$ for initialization. Hence, with aforementioned dueling structure in equation 7 and our pre-defined parameterization, one could set the advantage function as $A_{\theta_{\text{ckpt}}}(s,a) := \alpha[\ell_{\theta_{\text{ckpt}}}(s,a) - \max_a \ell_{\theta_{\text{ckpt}}}(s,a)]$, leading to $Q_\phi(s,a) := A_{\theta_{\text{ckpt}}}(s,a) + V_\phi(s)$. See also our forward pass graph defined in Figure 2b. In a nutshell, this $Q_\phi$-function produces a policy $\pi_\phi$ identical to the output distribution $p_{\theta_{\text{ckpt}}}$ of $\text{LM}_{\theta_{\text{ckpt}}}$,

$$\pi_\phi(a|s) = \text{softmax}[\tfrac{1}{\alpha}\mathbf{Q}_\phi(s)][a] = \text{softmax}[\boldsymbol{\ell}_{\theta_{\text{ckpt}}}(s) - \max_a \ell_{\theta_{\text{ckpt}}}(s,a) + \tfrac{1}{\alpha}V_\phi(s)][a] = p_{\theta_{\text{ckpt}}}(a|s), \qquad (11)$$

where $\mathbf{Q}(s) := [Q(s,a)]_{a \in \mathcal{A}}$ and $\boldsymbol{\ell}(s) := [\ell(s,a)]_{a \in \mathcal{A}}$.

Recalling equation 5 - 6, the $Q$-learning family can be viewed as iterations between policy evaluation and improvement. We now elaborate on how this $Q_\phi$ function initialization affects both steps.

**Policy improvement.** One could, informally, consider the operation of taking softmax with respect to $\frac{1}{\alpha}Q_\phi$ as a *soft policy improvement* (Haarnoja et al., 2018) step with a temperature $\alpha$. Therefore, equation 11 can be interpreted as: running soft policy improvement alone with this initialized $Q_\phi$ preserved the performance of pre-trained $\text{LM}_{\theta_{\text{ckpt}}}$, offering a good starting point of online fine-tuning.

**Policy evaluation.** Yet, this $Q_\phi$ function only captures relative values, since we initialized only the advantages $A_{\theta_{\text{ckpt}}}$ - the relative information - as shown in equation 11. $V_\phi$ can thereby be an arbitrary function. This would not affect the policy improvement step due to the translation invariance of the softmax function. However, during the policy evaluation step, see e.g. equation 5, the Bellman error can be heavily influenced by the $V$-values. When the $V$-values is the dominant source of error, the policy evaluation optimization could be largely driven by the *state-only* $V$-values. This can lead to a loss of the *relative action values*, that we intended to preserve in the previous step.

**Pre-training of $V_\phi$.** This can be addressed by adding a pre-training phase of $V_\phi(s)$, during which we *freeze* the advantage function $A_{\theta_{\text{ckpt}}}$ and train $V_\phi$ by minimizing the temporal difference error (or equivalently doing policy evaluation). In this stage, we optimize the following loss until convergence

$$\mathcal{L}_V(V_\phi; \ell_{\theta_{\text{ckpt}}}) = \frac{1}{T}\mathbb{E}_{(s_t,a_t,r_t,s_{t+1}) \sim \mathcal{D}_{\text{train}}}\sum_{t=0}^{T}\left[r_t + \gamma\,\text{SG}\left(Q_\phi(s_{t+1},\hat{a}_{t+1})\right) - Q_\phi(s_t,a_t)\right]^2, \quad (12)$$

where SG is a stop gradient operator, $\text{SG}(Q_\phi(s',\hat{a}'))$ follows standard semi-gradient optimization, $\hat{a}_{t+1}$ is a target action (details deferred to section 4.3), and $Q_\phi(s,a) = A_{\theta_{\text{ckpt}}}(s,a) + V_\phi(s)$.

In summary, our initialization steps ensures that, prior to fine-tuning $\theta$, our $Q_\phi$ meets two important conditions: it starts with the action distribution $p_{\theta_{\text{ckpt}}}$ of a pre-trained $\text{LM}_{\theta_{\text{ckpt}}}$, and it begins with low temporal difference error (because the pre-training of $V_\phi$ in equation 12 directly minimizes it).

## 4.3 A CONSERVATIVE BELLMAN OPERATOR

With a pre-trained state value function $V_\phi$, we are now ready to learn a good state-action value function via fine-tuning. We parameterize $Q_\theta(s,a) := A_\theta(s,a) + V_\theta(s) = \alpha[\ell_\theta(s,a) - \max_a \ell_\theta(s,a)] + V_\theta^r + V_\phi$, where we define $V_\theta = V_\theta^r + V_\phi$, and we initialize $\theta$ in a way such that $\ell_\theta = \ell_{\theta_{\text{ckpt}}}$ and $V_\theta^r = 0$. It ensures that $Q_\theta = Q_\phi$ on initialization, a good starting point for subsequent fine-tuning on $\theta$. Technically speaking, setting $V_\theta = V_\theta^r + V_\phi$ is not required, as one could finetune both $\theta$ and $\phi$. We however observed that finetuning a residual head $V_\theta^r$, with $\phi$ frozen, leads to better stability.

Although we avoid training $Q_\theta$ from scratch, optimizing $Q_\theta$ by $Q$-learning family algorithms can still be challenging. We attribute this to the characteristics of the Bellman optimality operator $\mathcal{B}^*$ that seeks to learn the optimal value function $Q^*$ and optimal policy $\pi^*$, which requires a good data coverage of the state-action space $\mathcal{S} \times \mathcal{A}$ (e.g. Jiang & Huang, 2020; Xie et al., 2021a; Zhan et al., 2022). In program synthesis, however, such assumption can hardly be met due to the large state-action space and the high computational costs of Transformer inference. While conventional $Q$-learning family relies on the operator $\mathcal{B}^*$, recent works in RL, especially those considering limited data regime (e.g. Agarwal et al., 2020; Levine et al., 2020), often design "conservative" operators (e.g. Achiam et al., 2017; Kumar et al., 2020; Brandfonbrener et al., 2021) to address difficulties led by $\mathcal{B}^*$.

**Conservative Bellman operators.** The concept behind conservative Bellman operators is to "aim low". Instead of learning the optimal $Q^*$ and $\pi^*$, these operators typically seeks to learn a policy $\pi$ that either surpasses a behavior policy (which is used to collect a RL dataset in offline RL literature, see e.g. Achiam et al., 2017; Brandfonbrener et al., 2021) or fine-tune a pre-existing policy (e.g. Xie et al., 2021b; Yu & Zhang, 2023). This is often achieved by introducing a regularizer that penalizes deviations from the behavior/pre-existing policy. In particular, as shown in equation 14, we define our conservative Bellman operator $\mathcal{B}^q$, which depends on a *fixed, pre-defined* policy $q$, as follows:

$$\text{optimality } \mathcal{B}: \quad (\mathcal{B}^*Q)(s,a) = r(s,a) + \gamma\mathbb{E}_{s'}[Q(s',\hat{a}')], \text{where } \hat{a}' = \arg\max_a Q(s',a) \quad (13)$$

$$\text{conservative } \mathcal{B}: \quad (\mathcal{B}^qQ)(s,a) = r(s,a) + \gamma\mathbb{E}_{s'}[Q(s',\hat{a}')], \text{where } \hat{a}' = \arg\max_a q(a|s') \quad (14)$$

The intuition behind our operator $\mathcal{B}^q$ is that we evaluate the action-value function $Q^{q^\uparrow}$ of a greedified policy $q^\uparrow(a|s) := \mathbb{1}\{a = \arg\max_a q(a|s)\}$, where $\mathbb{1}$ is the indicator function. The rationale

behind greedification is that $q^{\uparrow}$ can be seen as $q$ in a greedy-decoding mode, which usually has better (one-shot) capability than sampling mode (although the latter has better generation diversity). Considering setting $q = p_{\theta_{\text{ckpt}}}$, the operator $\mathcal{B}^{p_{\theta_{\text{ckpt}}}}$ seeks to learn a policy $\pi$ that outperforms $p_{\theta_{\text{ckpt}}}$.

We further comment on some properties of $\mathcal{B}^q$: proposition 4.1 shows $\mathcal{B}^q$ is a contraction, meaning there is an unique fixed point. It leads to proposition 4.2, motivating our development of Section 4.5.

**Proposition 4.1.** $\mathcal{B}^q$ *is $\gamma$-contraction in $\ell_\infty$ norm.*

Given our conservative Bellman operator, we could define our conservative temporal difference loss,

$$\mathcal{L}_Q(Q_\theta; q) = \tfrac{1}{T}\mathbb{E}_{(s_t,a_t,r_t,s_{t+1})\sim\mathcal{D}_{\text{train}}}\textstyle\sum_{t=0}^{T}\left[r_t + \gamma \,\text{SG}\left(Q_\theta(s_{t+1}, \hat{a}_{t+1})\right) - Q_\theta(s_t, a_t)\right]^2, \quad (15)$$

where $\hat{a}_{t+1} = \arg\max_a q(a|s_{t+1})$, and $Q_\theta(s,a) = \alpha\left[\ell_\theta(s,a) - \max_a \ell_\theta(s,a)\right] + V_\theta^r(s) + V_\phi(s)$.

## 4.4 IMPLEMENTATION AND OPTIMIZATION

**Architecture and parameterization recap.** Following (Le et al., 2022; Shojaee et al., 2023; Liu et al., 2023), we choose T5 (Raffel et al., 2020) as our base architecture for $\theta_{\text{ckpt}}$, $\phi$ and $\theta$; and $\theta_{\text{ckpt}}$ is initialized with CodeRL checkpoint which is publicly available. Specifically, $\theta_{\text{ckpt}}$, $\phi$ and $\theta$ share a same encoder, and the encoder is frozen throughout, to reduce the amount of learnable parameters.

**Two-stage training.** As noted earlier, our training are composed with two stages: a pre-training stage of $\phi$, namely $\phi$-stage, and a fine-tuning stage of $\theta$, namely $\theta$-stage. A pseudo-algorithm could be found in Appendix A. In addition, further implementation details are deferred to Appendix H.

$\phi$-stage: Given our development of Section 4.2, we pre-train $V_\phi$ function using stochastic gradient descent with $\nabla_\phi \mathcal{L}_V(V_\phi; \ell_{\theta_{\text{ckpt}}})$, with $\mathcal{L}_V$ defined in equation 12.

$\theta$-stage (fine-tuning): In this stage, we seek to optimize $Q_\theta$ to minimize our previously developed losses: $\mathcal{L}_{\text{adv}}$ and $\mathcal{L}_Q$, as defined in equation 8 and 15, respectively. In addition, it is also a common practice to include a cross-entropy loss $\mathcal{L}_{\text{ce}}$ during fine-tuning. Therefore, we conclude our final loss function as equation 17, and $\theta$ is updated using stochastic gradient descent with $\nabla_\theta \mathcal{L}_{\text{ft}}(Q_\theta; p_{\theta_{\text{ckpt}}})$.

$$\text{Recall: } Q_\theta(s,a) = A_\theta(s,a) + V_\theta(s) = \alpha\left(\ell_\theta(s,a) - \max_a \ell_\theta(s,a)\right) + V_\theta^r(s) + V_\phi(s) \quad (16)$$

$$\mathcal{L}_{\text{ft}}(Q_\theta; p_{\theta_{\text{ckpt}}}) = \mathcal{L}_Q(Q_\theta; p_{\theta_{\text{ckpt}}}) + \beta_{\text{adv}}\mathcal{L}_{\text{adv}}(A_\theta) + \beta_{\text{ce}}\mathcal{L}_{\text{ce}}(\pi_\theta), \text{ where } \pi_\theta = \text{softmax}\left(\tfrac{1}{\alpha}Q_\theta\right). \quad (17)$$

## 4.5 A FREE REWARD MODEL

Reward modeling is crucial in language modeling and also in inverse RL (detailed discussions could be found in Appendix C). An intriguing finding from IRL, applicable to our framework, is that a trained $Q$-function can recover a reward function without additional training. Analogously to Garg et al. (2021), an one-to-one correspondence between $Q$ and reward holds with our conservative Bellman operator $\mathcal{B}^q$. We define the inverse conservative Bellman operator $\mathcal{T}^q : \mathbb{R}^{\mathcal{S}\times\mathcal{A}} \to \mathbb{R}^{\mathcal{S}\times\mathcal{A}}$,

$$(\mathcal{T}^q Q)(s,a) = Q(s,a) - \gamma\mathbb{E}_{s'}Q\left(s', \arg\max_a q(a|s')\right). \quad (18)$$

**Proposition 4.2.** *The inverse conservative Bellman operator $\mathcal{T}^q$ is a bijection.*

Proposition 4.2 shows that a $Q_\theta$ is uniquely corresponding to a reward function $\tilde{r}_\theta := \mathcal{T}^q Q_\theta$.[1] Given the definition of $\mathcal{T}^q$ we could recover a reward model $\tilde{r}_\theta$ with $Q_\theta$ *without additional training*:

$$\tilde{r}_\theta(s,a) = Q_\theta(s,a) - \gamma\mathbb{E}_{s'}Q_\theta\left(s', \arg\max_a p_{\theta_{\text{ckpt}}}(a|s')\right) \approx Q_\theta(s,a) - \gamma V_\theta(s'). \quad (19)$$

We use the estimation $\tilde{r}_\theta(s,a) \approx Q_\theta(s,a) - \gamma V_\theta(s')$ in practice, with reasons deferred to Appendix F.

**Candidates selection with $\tilde{r}_\theta$.** We leverage our reward model $\tilde{r}_\theta$ to do candidate programs selection, as an example to highlight the additional benefits of value-based RL. We rank generated programs by the cumulative rewards $\tilde{R}_\theta(W) := \sum_{t=0}^{T}\tilde{r}_\theta(s_t, a_t)$, predicted by our reward model $\tilde{r}_\theta$, to select the programs that are most likely to be correct. Specifically, for pass@$k$ metrics, we follow the evaluation protocol used in CodeT (Chen et al., 2022), a work that considered program selection via automatic generated tests. This protocol computes pass@$k$ by first generating $m$ programs and select a subset of $k$ programs to evaluate pass@$k$. In our case, we select the $k$-sized subset with top-$k$ highest $\tilde{R}_\theta(\cdot)$ from total $m$ candidates. Our results in Section 5 follow this evaluation protocol.

---

[1] We use $\tilde{r}$ and $r$ to name our recovered reward model and the real reward function, respectively.

Table 1: Empirical evaluation on APPS test set. [†], [‡] and [‡‡] indicates results duplicated from Le et al. (2022), Shojaee et al. (2023) and Liu et al. (2023), respectively. Bold **number** indicates the best result and underlined number means our result are the second best. Intro, inter and comp stand for introductory, interview and competition, respectively.

| Model | # trainable parameters | Pass@1 | | | | Pass@5 | | | | Pass@1000 | | | |
|---|---|---|---|---|---|---|---|---|---|---|---|---|---|
| | | Intro | Inter | Comp | All | Intro | Inter | Comp | All | Intro | Inter | Comp | All |
| Codex[†] | 12B | 4.14 | 0.14 | 0.02 | 0.92 | 9.65 | 0.51 | 0.09 | 2.25 | 25.02 | 3.70 | 3.23 | 7.87 |
| AlphaCode[†] | 1B | - | - | - | - | - | - | - | - | 17.67 | 5.24 | 7.06 | 8.09 |
| GPT3[†] | 175B | 0.20 | 0.03 | 0.00 | 0.06 | - | - | - | - | - | - | - | - |
| GPT2[†] | 0.1B | 1.00 | 0.33 | 0.00 | 0.40 | 2.70 | 0.73 | 0.00 | 1.02 | - | - | - | - |
| GPT2[†] | 1.5B | 1.30 | 0.70 | 0.00 | 0.68 | 3.60 | 1.03 | 0.00 | 1.34 | 25.00 | 9.27 | 8.80 | 12.32 |
| GPT-Neo[†] | 2.7B | 3.90 | 0.57 | 0.00 | 1.12 | 5.50 | 0.80 | 0.00 | 1.58 | 27.90 | 9.83 | 11.40 | 13.76 |
| GPT-J[†] | 6B | 5.60 | 1.00 | 0.50 | 1.82 | 9.20 | 1.73 | 1.00 | 3.08 | 35.20 | 13.15 | 13.51 | 17.63 |
| RL based methods - without using example unit tests | | | | | | | | | | | | | |
| CodeRL[†] | 770M | 6.20 | **1.50** | 0.30 | 2.20 | 9.39 | 1.90 | 0.42 | 3.10 | 35.30 | 13.33 | 13.60 | 17.78 |
| PPOCoder[‡] | 770M | 5.20 | 1.00 | **0.50** | 1.74 | 9.10 | 2.50 | **1.20** | 3.56 | 35.20 | 13.35 | 13.90 | 17.77 |
| RLTF[‡‡] | 770M | 4.16 | 0.97 | 0.20 | 1.45 | 10.12 | **2.65** | 0.82 | 3.78 | **38.30** | **15.13** | **15.90** | **19.92** |
| $\mathcal{B}$-Coder | $\leq$770M/stage[3] | **6.70** | **1.50** | 0.30 | **2.30** | **10.40** | 2.63 | 0.70 | **3.80** | 37.00 | 13.67 | 12.60 | 18.12 |

**Remarks on $\tilde{r}_\theta$.** To further explain the motivation of ranking with $\tilde{r}_\theta$, consider a realistic deployment setting where a fine-tuned model is deployed for end-user applications. Users often provide a language description of their needs but may not include test cases (which can also be challenging for beginners or casual users). Additionally, the model is usually required to offer a single best response instead of a range of options. Therefore, the ability to rank programs without true rewards is a desirable advantage.

To preview the effectiveness of $\tilde{r}_\theta$, we show the correlation between environmental reward $r$ and our cumulative reward $\tilde{R}_\theta$. In Figure 3, green region corresponds to correct programs, and has the highest $\tilde{R}_\theta$ on average. For incorrect programs, those with compile and runtime errors have the lowest and the second lowest $\tilde{R}_\theta$, respectively. Programs can be executed but fail some tests, have the second

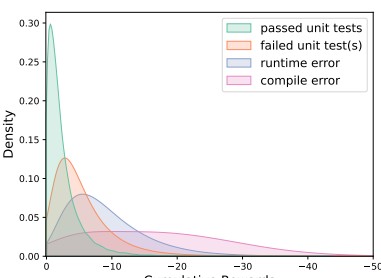

Figure 3: Kernel density estimation of $\tilde{R}_\theta(\cdot)$ evaluated on a collection of generated programs. The x-axis represents the predicted reward given by $\tilde{R}_\theta$ and the y-axis is its density. Color codes the true outcomes defined in equation 2.

highest $\tilde{R}_\theta$. Hence, it concludes that $\tilde{R}_\theta$ has an evident positive correlation to the true reward $r$.

## 5 EMPIRICAL EVALUATION

**Sampling using $Q_\theta$.** Nucleus sampling (top-$p$ sampling) (Holtzman et al., 2019) with sampling temperature[2] (Ackley et al., 1985) has been one of the most important sampling techniques. It can also be easily implemented in our framework. One could simply consider $Q_\theta/\alpha$ as logits and the sampling procedure would remain identical to standard LMs, see Appendix B for details.

**APPS benchmark and baselines.** In line with prior RL-based works (Le et al., 2022; Shojaee et al., 2023; Liu et al., 2023), we evaluate $\mathcal{B}$-Coder on the challenging code contests benchmark APPS (Hendrycks et al., 2021). It contains 5,000 training and 5,000 testing problems, with three difficulty levels: introductory, interview and competition. We compare our $\mathcal{B}$-Coder with pre-trained or supervise fine-tuned LLM baselines: GPT2 (Radford et al., 2019), GPT3 (Brown et al., 2020), GPT-Neo (Black et al., 2021), GPT-J (Wang & Komatsuzaki, 2021), Codex (Chen et al., 2021a) and AlphaCode (Li et al., 2022); and RL fine-tuned baselines: CodeRL (Le et al., 2022), PPOCoder (Shojaee et al., 2023) and a concurrent work RLTF (Liu et al., 2023).

**APPS: without example test outcomes.** In the APPS dataset, each problem has several example unit tests (different from the hidden unit tests used for evaluation). These example tests are usually leveraged to refine generated samples. For example, CodeRL and RLTF considers a critic sampling

---

[2]Sampling temperature is different from temperature $\alpha$ in equation 10. They can be different values.

[3]For both $\phi$ and $\theta$-stage, our model trains a decoder and heads, i.e. $\leq$770M trainable params per stage.

(CS) strategy that refines and repairs generated programs based on the execution outcomes of the example tests. We start with experiments results in which example test outcomes are not used (hence CodeRL and RLTF results in Table 1 are without CS). Table 1 shows that our $\mathcal{B}$-Coder has overall the best pass@$k$ for $k = \{1, 5\}$ and achieves second best place for $k = 1000$ (best result reported by the concurrent work RLTF). For Table 1 results, we use nucleus sampling with a sampling temperature of 0.6. We set $m$ to 256 for $k = \{1, 5\}$ and $m$ to 2500 for $k = 1000$, where $m$ is a hyper-parameter of our ranking protocol introduced in Section 4.5 (see Appendix I for an ablation study on $m$).

**APPS: using example test outcomes.** Table 2 lists the results using example tests. In addition to the CS strategy that uses example tests to refine/repair programs, Li et al. (2022) and Chen et al. (2021a) consider a *filtered setting*, in which programs failing example tests are excluded, and the pass@$k$ is evaluated us-

Table 2: APPS results when using example test outcomes.

| Model | Pass@1 | | | | Pass@5 | | | |
|---|---|---|---|---|---|---|---|---|
| | Intro | Inter | Comp | All | Intro | Inter | Comp | All |
| Codex[†] filtered | 22.78 | 2.64 | 3.04 | 6.75 | **24.52** | 3.23 | 3.08 | 7.46 |
| AlphaCode[†] filtered | - | - | - | - | 14.36 | 5.63 | 4.58 | 7.17 |
| CodeRL[†] cs | 6.77 | 1.80 | 0.69 | 2.57 | 15.27 | 4.48 | 2.36 | 6.21 |
| CodeRL[†] filtered | 16.27 | 6.00 | **4.27** | 7.71 | - | - | - | - |
| CodeRL[†] cs+filtered | 16.52 | 6.16 | 4.15 | 7.83 | 24.49 | 8.58 | **7.82** | **11.61** |
| RLTF[‡‡] cs | 8.40 | 2.28 | 1.10 | 3.27 | 18.60 | 5.57 | 3.70 | 7.80 |
| $\mathcal{B}$-Coder filtered | **18.00** | **6.63** | 2.30 | **8.04** | 23.30 | **8.83** | 6.40 | 11.30 |

ing (a subset of) programs that pass example tests (which is also related to the $k@m$ metric (Li et al., 2022), the pass rate using $k$ submissions from $m$ samples). We also test $\mathcal{B}$-Coder in this filtered setting. Similarly, we first exclude programs that fail example tests. Suppose $n$ out of $m$ programs pass; we then follow our ranking protocol to get top-$k$ out of $n$ programs for evaluation. $\mathcal{B}$-Coder outperforms baselines with either CS or filtered setting for $k = \{1, 5\}$. The baseline, CodeRL+CS+filtered, incorporated both strategies achieved a slight advantage over $\mathcal{B}$-Coder for pass@5 while being surpassed by $\mathcal{B}$-Coder for pass@1. It worth mentioning that CS is a plug-and-play component, which could also be combined with $\mathcal{B}$-Coder, to further improve pass rate. For the results in Table 2, we use a temperature of 0.4 and $m$ set to 1000, matching the $m$ used in Le et al. (2022).

**Generalization ability.** In addition, we test the generalization ability of our dual strategy, ranking with $\tilde{R}_\theta$. We study two aspects: generalization to other models and generalization to different domains. To this end, we designed the following experiments, which confirmed its generalizability in positive.

For the former, we generate (off-policy) programs using CodeRL (with $m = 256$), and rank those programs by $\tilde{R}_\theta$. Table 3 shows our ranking strategy leads to improvements in most cases, even though the programs to be ranked are not generated by $\mathcal{B}$-Coder.

For the latter, we test our dual strategy with another dataset MBPP (Austin et al., 2021) (with $m = 512$). Table 4 shows consistent improvements for all temperatures and $k$.

Table 3: Generalization to CodeRL. Pass@$k$ evaluated with top-$k$ ranked programs, generated by CodeRL. $\cdot$ indicates absolute improvement achieved by ranking, compared to un-ranked pass@$k$.

| k | Temp. | Pass@k | | | | | | | |
|---|---|---|---|---|---|---|---|---|---|
| | | Intro | | Inter | | Comp | | All | |
| 1 | 0.4 | 6.30 | 1.91 | 1.27 | 0.37 | 0.50 | 0.37 | 2.12 | 0.68 |
| | 0.6 | 6.00 | 2.13 | 1.23 | 0.42 | 0.50 | 0.36 | 2.04 | 0.75 |
| 5 | 0.4 | 9.30 | -0.2 | 2.10 | 0.01 | 0.70 | 0.15 | 3.26 | 0.00 |
| | 0.6 | 10.20 | 0.58 | 2.57 | 0.41 | 0.80 | 0.16 | 3.74 | 0.39 |

Table 4: Zero-shot pass@$k$ on MBPP. $\cdot$ indicates absolute improvement achieved by ranking.

| Temp. | k=1 | | k=5 | | k=10 | | k=80 | |
|---|---|---|---|---|---|---|---|---|
| 0.7 | 20.13 | 6.61 | 37.04 | 5.61 | 44.45 | 4.63 | 64.00 | 1.41 |
| 0.8 | 18.89 | 6.99 | 36.59 | 7.21 | 44.46 | 6.59 | 65.20 | 4.28 |
| 0.9 | 17.32 | 7.34 | 35.04 | 8.58 | 43.15 | 8.22 | 63.20 | 4.33 |

## 6 CONCLUSION

In this work, we explore the feasibility of value-based RL algorithms for program synthesis task. We demonstrate how to stabilize and accelerate training through $Q$-function initialization and conservative updates. Moreover, our work is conducted with minimal reward engineering effort, thereby placing an emphasis on the perspective of algorithm designs. While policy-based algorithms remain mainstream in the current program synthesis literature, the question of how to effectively leverage off-policy programs, including historical synthetic samples, in a principled way, might still be under-explored. We are convinced that value-based RL offers a promising direction to address this question, and thereby to scale RL for code generation at large by (re)-using the extensive collection of off-policy programs. Our work could thus serve as an important initial step towards this direction.

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

## A  PSEUDO-CODE FOR TRAINING

---

**Algorithm 1** Training Procedure with $\phi$- and $\theta$-stages

---

**Require:** $\theta_{\text{ckpt}}$, $\phi$, and $\theta$ with a shared frozen encoder
 1:  # pre-training stage, update $\phi$ only
 2:  **procedure** PRETRAIN$V$VALUE($\phi$)                                                                       ▷ $\phi$-stage
 3:      **for** num_iters **do**
 4:          Draw sample $(s, a, r, s')$ from dataset
 5:          Compute logits $\ell_{\theta_{\text{ckpt}}}(s, \cdot)$
 6:          Compute state value $V_\phi(s)$
 7:          Compute loss $\mathcal{L}_V(V_\phi; \ell_{\theta_{\text{ckpt}}})$                                    ▷ arguments omitted for brevity
 8:          Gradient step with $\nabla_\phi \mathcal{L}_V(V_\phi; \ell_{\theta_{\text{ckpt}}})$                  ▷ equation 12
 9:      **end for**
10:  **end procedure**
11:  # fine-tuning stage, update $\theta$ only
12:  **procedure** FINETUNE$Q$VALUE($\theta$)                                                                     ▷ $\theta$-stage
13:      **for** num_iters **do**
14:          Draw sample $(s, a, r, s')$ from dataset
15:          Compute residual state-value $V_\theta^r(s)$
16:          Compute pre-trained state-value $V_\phi(s)$
17:          Compute state-value $V_\theta(s) = V_\theta^r(s) + V_\phi(s)$
18:          Compute advantage $A_\theta(s, \cdot) = \ell_\theta(s, \cdot) - \max_a \ell_\theta(s, a)$
19:          Compute $Q_\theta(s, \cdot) = \alpha A_\theta(s, \cdot) + V_\theta(s)$                               ▷ equation 16
20:          Compute $\pi_\theta(\cdot|s) = \text{softmax}(Q_\theta(s, \cdot)/\alpha)$
21:          Compute $p_{\theta_{\text{ckpt}}}(\cdot|s)$                                                          ▷ equation 14
22:          Compute $\mathcal{L}_Q(Q_\theta; p_{\theta_{\text{ckpt}}})$                                          ▷ equation 15
23:          Compute $\mathcal{L}_{\text{ce}}(\pi_\theta)$ and $\mathcal{L}_{\text{adv}}(A_\theta)$               ▷ equation 1 and 8
24:          Compute fine-tune loss $\mathcal{L}_{\text{ft}}(Q_\theta; p_{\theta_{\text{ckpt}}}) = \mathcal{L}_Q(Q_\theta; p_{\theta_{\text{ckpt}}}) + \beta_{\text{ce}}\mathcal{L}_{\text{ce}}(\pi_\theta) + \beta_{\text{adv}}\mathcal{L}_{\text{adv}}(A_\theta)$
25:          Gradient step with $\nabla_\theta \mathcal{L}_{\text{ft}}(Q_\theta; p_{\theta_{\text{ckpt}}})$
26:      **end for**
27:  **end procedure**

---

## B  PSEUDO-CODE FOR SAMPLING

---

**Algorithm 2** Sampling Procedure

---

**Require:** model parameters $\theta$, $\phi$; SAMPLER$_{p,t}(\cdot) : \mathbb{R}^{1 \times |\mathcal{V}|} \to \mathcal{V}$ that maps a logits vector to a token
    with hyper-parameters $p$ (top-$p$ sampling) and temperature $t$
 1:
 2:  **procedure** SAMPLEONETOKEN($s$)
 3:      Obtain current state $s$
 4:      Compute logits vector $\boldsymbol{\ell}_\theta(s) \in \mathbb{R}^{1 \times |\mathcal{V}|}$
 5:      Compute advantage vector $\mathbf{A}_\theta(s) = \boldsymbol{\ell}_\theta(s) - \max_a \boldsymbol{\ell}_\theta(s)[a]$
 6:      Compute $V_\theta(s) = V_\theta^r(s) + V_\phi(s)$
 7:      Compute $Q$ vector $\mathbf{Q}_\theta(s) = \alpha \mathbf{A}_\theta(s) + V_\theta(s)$
 8:      Run SAMPLER$_{p,t}(\mathbf{Q}_\theta(s)/\alpha)$                                                          ▷ sample with $\mathbf{Q}_\theta(s)/\alpha$
 9:  **end procedure**

---

## C ADDITIONAL RELATED WORKS

**Off-policy policy-based methods.** One string of off-policy policy-based methods is based on importance ratio. Suppose the data is collected by a behavior policy $\beta$, PG with off-policy data can be corrected by $\nabla_\mu J(\mu) = \mathbb{E}_\beta[\frac{\pi_\mu(a_t|s_t)}{\beta(a_t|s_t)}(\sum_{i=t}^{T}\gamma^{i-t}r_i)\nabla_\mu \log \pi_\mu(a_t|s_t)]$. This allows unbiased gradient even though the data distribution is off-policy. However, computing the ratio $\pi_\mu(a|s)/\beta(a|s)$ is not always feasible as the density function of off-policy data, such as human data, is often unknown. In addition, this correction can lead to high variance due to the product of ratios along trajectories.

While vanilla importance-weighted off-policy PG does not require the approximation of value functions, some advanced ratio-based methods often incorporate value functions, such as (Imani et al., 2018; Liu et al., 2020). Another viable approach is the direct combination of value-based and policy-based methods, often referred to as the actor-critic framework, e.g. (Konda & Tsitsiklis, 1999; Degris et al., 2012). Although actor-critic methods are often conisdered as the third category, besides policy-based and value-based, we and some other works (Fujimoto et al., 2018) lean towards categorizing actor-critic to be more value-based, as the major difficulty lies in value function approximations. Nevertheless, both directions of extending policy-based methods to an off-policy setting, largely rely on the value functions. This emphasizes the motivation and significance of our work.

**Reward modeling and beyond.** Due to the successes of reinforcement learning from human/AI feedback (Christiano et al., 2017; Bai et al., 2022b). Reward modeling and RL fine-tuning with learned reward model has been a popular choice for post-SFT (supervised fine-tuning) refinement (see e.g. Ziegler et al., 2019; Stiennon et al., 2020; Bai et al., 2022a; Ouyang et al., 2022). In particular, in program synthesis, Le et al. (2022) trains a classifier, that predicts unit test outcomes, as their reward model for RL fine-tuning. However, reward models can sometimes be expensive to train and their quality can heavily impact RL fine-tuning performance. Recent works (e.g. Rafailov et al., 2023; Diao et al., 2023) explore preference learning beyond conventional reward model.

Modeling reward function, on the other hand, has been a long-lasting topic in inverse RL or imitation learning (IRL or IL, see e.g. Ng et al., 2000; Abbeel & Ng, 2004; Ziebart et al., 2008; Ho & Ermon, 2016). While conventional IRL/IL often iterates between reward model fitting and RL training stages, recent IL works (Jacq et al., 2019; Garg et al., 2021) also explore beyond explicitly reward modeling to reduce training instability and optimization difficulty, led by the iterative optimization scheme. Specifically, Garg et al. (2021) leverages the one-to-one correspondence between $Q$-function and reward model, given the soft Bellman operator, to eliminate the reward fitting step.

**Candidate selection in program synthesis.** Existing works have shown one could improve program pass rate by filtering out programs that are likely to be incorrect. For instance, Chen et al. (2021a) filtered out programs that cannot pass example unit tests given in doc-strings, and Chen et al. (2022) filtered out programs that cannot pass generated unit tests. Furthermore, reward models are also often used to rank candidate programs (see e.g. Gulcehre et al., 2023; Touvron et al., 2023).

## D A SPECTRUM OF RL APPLICATIONS

To conceptually demonstrate the differences between policy-based and value-based methods, and why program synthesis might be well-suited to value-based approaches, Figure 4 presents a spectrum of RL applications. It could be observed that in scenarios where rewards are not expensive to evaluate or there's plenty of off-policy data (data not generated by the *current* policy/model) value-based methods tend to be preferred. Consider, for instance, InstructGPT (Ouyang et al., 2022) (policy-based) and AlphaGo (Silver et al., 2016) (value-based). The former relies on human annotators (*expensive*) to label model-generated (*on-policy*) responses, while the latter obtains rewards from simulators (*cheap*), and leverages (1) human expert games (*off-policy*) during training and (2) re-using historical games (*off-policy*) through experience reply.

Table 5 provides explanations our application plot of Figure 4. Applications in games typically find it easy to obtain rewards and make extensive use of off-policy data, e.g human games or historical replays. Conversely, InstructGPT obtains its rewards from preferences labeled by human annotators, with the data predominantly generated by the GPT model itself. The self-driving application notable has high cost of gathering rewards, due to the risks of real-world driving. While existing driving data

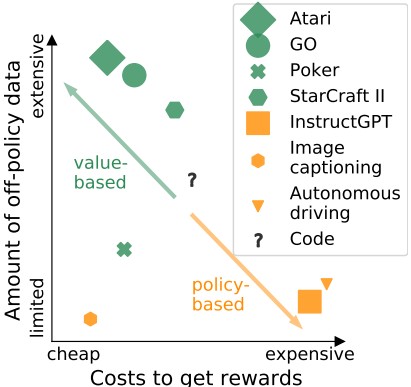

Figure 4: A collection of RL applications. ■ and ■ represents value-based and policy-based RL, respectively. The x-axis shows the difficulty of obtaining rewards, while the y-axis measures the amount of off-policy data. Tasks that face significant hurdles in gathering rewards or have limited off-policy data typically lean towards policy-based algorithms. Tasks where rewards are more readily obtained or that benefit from a substantial collection of off-policy data favors value-based methods. See descriptions of each task in Table 5.

could be utilized, Kendall et al. (2019) specifically choose not to use pre-collected data, leading to their choice of a policy-based algorithm.

In code generation, despite the availability of cheap rewards and the existing collection of off-policy programs, whether human-written or historical synthetic programs, current literature leans towards policy-based methods. We believe that value-based methods could be a promising direction, given their similarity to tasks with simulators.

Table 5: Summary of RL applications.

|  | References | Type of RL | Costs of Getting Rewards | Available Off-Policy Data |
|---|---|---|---|---|
| Atari | (Mnih et al., 2013) | value | cheap: simulator | extensive: history/human games |
| GO | (Silver et al., 2016) | value | cheap: simulator | extensive: history/human games |
| Poker | (Moravčík et al., 2017) (Brown & Sandholm, 2018) | value[4] | cheap: simulator | extensive: history/human games |
| StarCraft II | (Arulkumaran et al., 2019) | value | cheap: simulator | extensive: history/human games |
| InstructGPT | (Ouyang et al., 2022) | policy | expensive: human annotators | limited: mostly model-generated data |
| Image Caption | (Ranzato et al., 2015) (Rennie et al., 2017) | policy | cheap: automatic metrics | limited: mostly model-generated data |
| Self-driving | (Kendall et al., 2019) | policy | expensive: driving in real-world | limited: mostly model-generated data |
| Code Generation | (Le et al., 2022) (Shojaee et al., 2023) (Liu et al., 2023) | policy | cheap: unit testing | extensive: collection of human programs |

## E    REWARD ENGINEERING COMPARISON

Table 6 shows that ours has the least reward engineering effort. Note that our reward model $\tilde{r}_\theta$ is directly derived from $Q_\theta$, and is not used for training.

Table 7 shows the results when only basic reward function (defined in equation 2) is used, under no example test outcomes setting. CodeRL and RLTF results are duplicated from their reports.

---

[4]While Poker AI often uses counterfactual regret minimization (Zinkevich et al., 2007), which isn't strictly reinforcement learning, the shared principle of estimating action values allows us to categorize it under value-based methods.

Table 6: Comparison of reward designs

| Reward | Remark | Ours | CodeRL | RLTF | PPOCoder |
|---|---|---|---|---|---|
| Basic | equation 2 | ✓ | ✓ | ✓ | ✓ |
| Reward Model | learned reward model | | ✓ | | |
| Fine-Grained | fine-grained error type & location of error | | | ✓ | |
| Adaptive | ratio of passed tests | | | ✓ | |
| Syntactic Correctness | compilable | | | | ✓ |
| Syntactic Matching | syntactic similarity to ground truth | | | | ✓ |
| Semantic Matching | semantic similarity to ground truth | | | | ✓ |

Table 7: Performance with only basic reward (equation 2). $^{\dagger}$and $^{\ddagger\ddagger}$ indicates results duplicated from Le et al. (2022) and Liu et al. (2023), respectively.

| Model | Pass@1 | | | | Pass@5 | | | |
|---|---|---|---|---|---|---|---|---|
| | Intro | Inter | Comp | All | Intro | Inter | Comp | All |
| CodeRL$^{\dagger}$ | 4.60 | 1.10 | 0.20 | 1.62 | 7.10 | 1.57 | 0.40 | 2.44 |
| RLTF$^{\ddagger\ddagger}$ | - | - | - | 1.37 | - | - | - | 3.50 |
| $\mathcal{B}$-Coder | **6.70** | **1.50** | **0.30** | **2.30** | **10.40** | **2.63** | **0.70** | **3.80** |

# F  ADVANTAGE OF APPROXIMATE VERSION OF $\tilde{r}$

Recap that our recovered reward $\tilde{r}$ is computed by

$$\tilde{r}_\theta(s,a) = Q_\theta(s,a) - \gamma\mathbb{E}_{s'}Q_\theta\left(s', \arg\max_a p_{\theta_{\text{ckpt}}}(a|s')\right) \approx Q_\theta(s,a) - \gamma V_\theta(s'). \quad (20)$$

Imagining a scenario in which we sample/decode using a trained $Q_\theta$, the forward pass will compute $Q_\theta(s,a)$ and $V_\theta(s)$ for each timestep, because of our dueling architecture. But $p_{\theta_{\text{ckpt}}}$ will not be evaluated during generation, because $p_{\theta_{\text{ckpt}}}$ is only used when computing $\mathcal{L}_Q(\,\cdot\,; p_{\theta_{\text{ckpt}}})$. Computing the exact version $Q_\theta(s,a) - \gamma\mathbb{E}_{s'}Q_\theta(s', \arg\max_a p_{\theta_{\text{ckpt}}}(a|s'))$ will require additional computation of $p_{\theta_{\text{ckpt}}}$ during generation. In contrast, $Q(s,a)$ and $V(s)$ are already computed during generation, therefore it requires almost no additional computation to compute $\tilde{r}_\theta(s,a)$.

# G  ADDITIONAL IMPLEMENTATION TRICKS

## G.1  UPPER BOUND OF $Q$-FUNCTION

Given our reward design in equation 2, the cumulative reward is upper bounded by $R_{\max} = 1$. We enforce $Q(s,a) \leq R_{\max}$ by transform the state value function as $V(s) = -\text{SOFTABS}\,(V(s)) + R_{\max} \leq R_{\max}$, where $\text{SOFTABS}(x) := [\text{SOFTPLUS}(x) + \text{SOFTPLUS}(-x)]/2 + \ln 2$ is a soft absolute function. Given $A(s,a) \leq 0$, enforcing $V(s) \leq R_{\max}$ leads to $Q(s,a) \leq R_{\max}$.

## G.2  RESIDUAL HEAD INITIALIZATION

In section 4.3, we initialize $\theta$ in a way such that $\ell_\theta = \ell_{\theta_{\text{ckpt}}}$ and $V_\theta^r(s) = 0$. The former can be done by simply loading the checkpoint $\theta_{\text{ckpt}}$. Adding a residual head $V_\theta^r$, that initialized to output zeros, can be done with a simple trick. One can simply add two heads $h_1$ and $h_2$, let $h_1$ be trainable, and $h_2$ be fixed for subsequent fine-tuning, setting $V_\theta^r = h_1 - h_2$ achieves the desired functionality.

# H  TRAINING AND EVALUATION DETAILS

In supplement to implementation details in Section 4.4 and 5, we give more low-level details here.

**APPS dataset.** In addition to the train/test split details described in Section 5, APPS datast, on average, consists of 2 example unit tests, 21 hidden unit tests, and 23 ground truth programs. We

follow the same procudure as Hendrycks et al. (2021); Le et al. (2022) to construct prompts for both training and evaluation. Specifically, see Section 3 of Hendrycks et al. (2021).

**MBPP dataset.** MBPP has 974 instances with a 374/90/500 train/val/test splits and, in addition, 10 problems reserved for few-shot learning. Because we only do zero-shot evaluation on MBPP, only the 500 test problems are used for evaluation. Each problem of MBPP usually comes with three unit tests. In addition, these tests are usually not hidden. Therefore, prior works Le et al. (2022); Shojaee et al. (2023); Liu et al. (2023) often explicitly incorporate the tests into prompt string. We follow WizardCoder (Luo et al., 2023) to construct our input format. Details could be found in this repo.

**Pre-trained model.** We initialize our model with CodeRL checkpoint publicly available at here, meaning we initialize $\theta_{\text{ckpt}}$, $\phi$, and $\theta$ from it. Note that we freeze encoder for both $\phi$-stage and $\theta$-stage, therefore the encoder is shared during both training and generation. For both training and generation, we set the maximum length to 600 and 512 for source and target sequences, respectively.

**Training data preparation.** While we use $\mathcal{D}_{\text{train}}$ to represent our training dataset, yet we have not elaborated on how it is constructed. In general, we follow the protocol of prior RL-based works that combining all ground truth programs and a set of programs generated by the pre-trained model, for each problem $D$. Specifically, we generate 256 programs per problem using pre-trained checkpoint. Combined with ground truth programs, there are, on average, 278 programs per problem.

**Mini-batch preparation.** By prior definition, our dataset $\mathcal{D}_{\text{train}}$ now contains both ground truth programs and generated programs. Notably, the volume of generated programs is significantly larger than that of the ground truth programs. This means that if one were to randomly sample from the dataset, generated programs would dominate the mini-batches. To address this, when preparing a mini-batch, we sample $\rho_{\text{real}} \times B$ ground truth programs and $(1 - \rho_{\text{real}}) \times B$ generated programs, where $B$ is batch size.

**$\phi$-stage training.** In the $\phi$-stage, we pre-train state-value function $V_\phi(s)$. We conduct our experiment with $4\times$A100-80G GPUs. Specifically, we use batch size of 16 for each GPU and gradient accumulation step of 4, resulting in a total batch size of 256. For optimizer and scheduler, we use AdamW optimizer (Loshchilov & Hutter, 2018) with a constant learning rate of 1e-5 and a weight decay of 0.05. We train $\phi$ for 18k gradient steps.

**$\theta$-stage training.** In the $\theta$-stage, we conduct our experiment with $8\times$A100-80G GPUs. Specificaly we use batch size of 16 for each GPU and gradient accumulation step of 1, resulting in a total batch size of 128. For optimizer and scheduler, we use AdamW with a peak learning rate 3e-5, a weight decay of 0.05, and a linear decay scheduler with no warmup. We train $\theta$ for 10k gradient steps.

**Other hyper-parameters.** We set the ground truth data ratio $\rho_{\text{real}} = 0.5$ and the energy-based policy temperature $\alpha = 1$ (see equation 10) for all experiments. In $\theta$-stage, we use $\beta_{\text{adv}} = 0.1$ and $\beta_{\text{ce}} = 0.5$ .

## I  ABLATION ON $m$

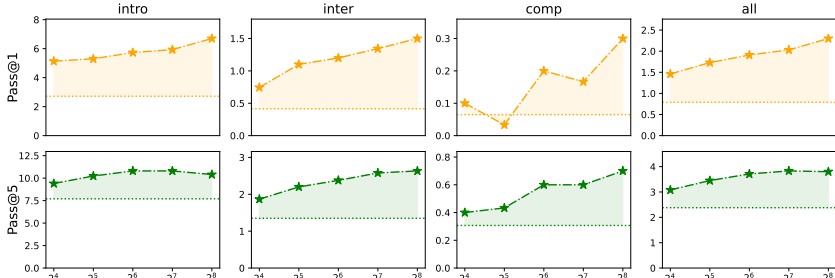

Figure 5: Ablation on $m$: our ranking strategy achieves consistent improvements under different budgets $m$.

Table 5 conduct an ablation study on ranking budgets $m$, it can be observed that our ranking strategy achieves consistent improvements under different budgets $m$.

## J    COMMENTS ON $\mathcal{B}^q$ PROPERTIES

### J.1    PROPOSITION 4.1

*Proof.*

$$\|\mathcal{B}^q Q_1 - \mathcal{B}^q Q_2\|_\infty = \max_{s,a} |r(s,a) + \gamma \mathbb{E}_{s'} Q_1(s',\hat{a}') - r(s,a) - \gamma \mathbb{E}_{s'} Q_2(s',\hat{a}')|$$

$$(\hat{a}' = \arg\max_a q(a|s'))$$

$$= \max_{s,a} \gamma \left| \mathbb{E}_{s'} [Q_1(s',\hat{a}') - Q_2(s',\hat{a}')] \right| \tag{21}$$

$$\leq \max_{s,a} \gamma \mathbb{E}_{s'} |Q_1(s',\hat{a}') - Q_2(s',\hat{a}')| \tag{22}$$

$$\leq \max_{s,a} \gamma \mathbb{E}_{s'} \max_{s',a'} |Q_1(s',a') - Q_2(s',a')| \tag{23}$$

$$= \gamma \|Q_1 - Q_2\|_\infty \tag{24}$$

$\square$

### J.2    PROPOSITION 4.2

*Proof.* The proof is similar to Lemma C.3. in Garg et al. (2021). To prove that $\mathcal{T}^p$ is a bijection, it suffices to show that for any $r : \mathcal{S} \times \mathcal{A} \to \mathbb{R}$, there exists a unique $Q : \mathcal{S} \times \mathcal{A} \to \mathbb{R}$ such that $r = \mathcal{T}^p Q$. Note that by proposition 4.1, there exists a unique $Q^p = \mathcal{B}^p r$ that satisfies $Q^p(s,a) = r(s,a) + \gamma \mathbb{E}_{s'} Q^p(s', \arg\max_a p(a|s'))$. Rearranging the terms gives $r = \mathcal{T}^p Q^p$. This completes the proof. $\square$

## K    DISCUSSION ON LIMITATIONS

Table 8:   Pass@1 results are evaluated with greedy decoded programs, and pass@$\{5, 50, 100\}$ are computed by sampled programs using a temperature of 0.4.

| Pass@ | CodeRL | $\mathcal{B}$-Coder |
|---|---|---|
| 1 | **1.60** | **1.60** |
| 5 | **3.28** | 2.88 |
| 50 | 7.16 | **7.35** |
| 100 | 8.76 | **9.18** |

Table 9: Ranking with $\tilde{r}$ compared with filtering with real environmental reward function $r$, i.e.   hidden tests. $\tilde{r}$-ranked results are duplicated from Table 1.

| | $\tilde{r}$-ranked | | $r$-filtered |
|---|---|---|---|
| | pass@1 | pass@5 | |
| Intro | 6.70 | 10.40 | 26.60 |
| Inter | 1.50 | 2.63 | 7.87 |
| Comp | 0.30 | 0.70 | 5.10 |
| All | 2.30 | 3.80 | 11.06 |

While being exploratory, our work admits certain limitations including: additional frozen parameters introduced, and we observe that raw performance (without ranking) is mixed compared to CodeRL (see Table 8) (which we believe is somewhat excusable as we use less reward designs). However, we remark the effectiveness of our overall framework including the dual strategy is non-trivial, especially with limited reward engineering.

It is also informative to show results filtered by the true environmental reward function $r$, instead of results ranked by our recovered reward function $\tilde{r}$. Although filtering with $r$ requires using hidden tests, meaning it cannot be implemented in realistic settings, also see discussions in Section 4.5. However, it could serve as an upper limit for our ranking strategy and as a sanity check. (Roughly speaking, if $\tilde{r} = r$, the pass rate of $\tilde{r}$-ranking and $r$-filtering would be identical.) To this end, we use the same set of candidate programs as those in Table 1, but apply the ground truth reward function $r$ to filter candidates rather than using $\tilde{r}$ for ranking. The corresponding results in Table 9 show that, although $\tilde{r}$-ranking is effective, there remains a large room for improvement.

