# OpenReview forum: "$\mathcal{B}$-Coder: Value-Based Deep Reinforcement Learning for Program Synthesis"
_ICLR.cc/2024/Conference — ICLR 2024 spotlight_

### Official Review · Reviewer_3d3T · 2023-10-27

**Soundness:** 3 good
**Presentation:** 3 good
**Contribution:** 3 good
**Rating:** 8
**Confidence:** 4

**Summary:**

In this paper, the authors introduce a value-based method for code generation with pretrained LLMs. In their opinion, their method fills a vacuum, due to the fact that all present LLM-based code-generation systems are policy-based in order to easily leverage pre-trained LLMs. The authors get around this difficulty via proposing initialisation and semi-gradient policy-improvement schemes in which the LLMs logits are visualised as Q-values. In addition to this, they propose a conservative Bellman operator in which the "best action" is not chosen based on the trained Q-values, but on the logits of the initial pre-trained LLM, which in their view stabilises training and alleviates the usual convergence problems of value-based RL.
The authors test their framework on a standard code-generation benchmark (the APPS benchmark) and use a relatively straightforward reward design. They report performance superior to, or competitive with, with an exhaustive set of baselines.

**Strengths:**

- The authors are correct in that value-based methods for code-generation were absent from the literature, and that the gap should be filled.
- Their solution for doing so is quite novel and elegant, especially the idea of visualising the LLM's logits as Q-values, and use them for both initialisation and policy improvement.
- They consider an exhaustive set of baselines.
- They show performance superior to, or competitive with, baselines, and on a programming benchmark notorious for its high difficulty.

**Weaknesses:**

- While the framework is novel, its comparison with baselines gives mixed results: in some cases, it does not actually come out on top. This contradicts the authors' claim that value-based method should be more suited to code generation: all of the baselines are policy-based.
- The authors filter programs at evaluation time using a different reward than the one used during training. This makes it hard to interpret their performance numbers. Could it be the filtering that is mostly responsible for them?
- The authors' method needs to be applied to a pre-trained LLM, which is used not only for initialisation but also for training. This is only possible with LLMs whose checkpoints are publicly available. In the case of the largest LLMs available today, even with a checkpoint available, considerable computational resources would be needed to apply the method.
- The paper has a bit of an idiosyncratic structure, with e.g. a whole section dedicated only to the reward filtering at evaluation time, a very small Related Work section, and no outline of the paper's contents in the introduction, which makes it a bit hard to read.
- Some more proof-reading is needed as typos can be found here and there.

**Questions:**

Main question and concern: why do the authors use a filtering procedure at evaluation time based on a different reward model $\tilde{r}(s,a)$ than the environmental reward $r(s,a)$ they use for training? Given that the real reward $r$ can be easily and cheaply computed (a point which the authors themselves make in the introduction), I cannot see the need for such a procedure, and the authors do not elaborate on it. The authors should either:
 - Justify this choice.
 - Recompute their performance numbers with filtering based on $r$ at evaluation time.
 - Re-train their method with the modelled reward $\tilde{r}$ from the outset.

Other concerns/questions/recommendations:
- Could the mixed performance numbers be due to the conservative Bellman operator, which keeps the policy close to the initial checkpoint? Did the authors consider alternative definitions for $q(a|s)$, or to update it at some point during training?
 - Value-based methods, including DQN, can suffer from the so-called "Deadly Triad" of deep RL. Can the authors elaborate on how they avoid this pitfall?
- It seems that the authors add a value head $V^r_\theta(s)$ (besides its LM head) to their LLM. How is that done, and can it be done with a generic LLM?
- In figure 2, it seems that the pass@1 metric is actually at its best at very beginning of training and that it never recovers afterwards. Is the figure only meant to be illustrative, or are those the real training curves?
- Can the authors outline their paper's content and structure at the end of the intro?
- In figure 3, it would be helpful to explicitly define what $\ell$ and $p$ stand for.
- Why is the Nucleus Sampling outlined at the end of section 3? In my opinion it would make more sense to have in section 5, since it appears to be part of the evaluation protocol. Or is it also used during training?
- It would be helpful if the authors defined (e.g. in a caption) what "Intro", "Inter", "Comp", and "All", mean in tables 1 and 2. While the text does mention it, it would be helpful to readers to be more explicit.
- Why do the authors not consider $k=1000$ in table 2, as they do in table 1?
- The study on generalisation of the reward model (notwithstanding my concerns on its use, detailed above) would better be moved to an appendix.

# Post rebuttal edit:
The authors have exhaustively addressed my concerns and revised their paper as I suggested. I shall therefore raise my score to 8.

**Details Of Ethics Concerns:**

No concerns.

---

> ### Author Response · Authors · 2023-11-21
> **Response to reviewer 3d3T [Part 1/3]**
>
> Thank you very much for your time in reviewing our work and your valuable comments! We are glad that you found our approach novel and elegant! We are enthusiastic about improving our paper by incorporating your constructive suggestions.
>
> ### Weaknesses and Main Question
>
> > its comparison with baselines gives mixed results: in some cases, it does not actually come out on top. This contradicts the authors' claim that value-based method should be more suited to code generation
>
> We believe the mixed results are largely influenced by the degree of reward engineering involved. Our approach intentionally uses **minimal reward engineering**, as shown in our Eq. (2), while the baselines use quite complex reward designs. A comparison is available in Table 5, Appendix C. Ablation studies in baseline papers, such as Table 2 in CodeRL, Figure 3(a) in PPOCoder, and Table 5 in RLTF, demonstrate that their reward designs contribute significantly to the performance.
>
> We were restraint on reward engineering because we seek to offer a clearer discussion on the design of value-based methods without blending in too much reward factors. This trade-off was made to enhance our focus on algorithmic insights. As the reviewer has highlighted, the fact that all baseline methods are policy-based adds value to our initial attempt of value-based training.
>
> > Main question and concern: why do the authors use a filtering procedure at evaluation time based on a different reward model $\tilde{r}(s, a)$ than the environmental reward $r(s, a)$ they use for training? Given that the real reward $r$ can be easily and cheaply computed
>
> We would like to response to the main question first, to clarify our choice of a different reward function $\tilde{r}(s, a)$ instead of the ground truth reward.
>
> The primary reason is the **availability of test cases**, which provide true rewards. During model training or evaluation, test cases are often available and required, as both training and test datasets are typically constructed with them. This context is what we meant by `real reward $r$ can be easily and cheaply computed'. However, **in practical scenarios, test cases are not always available**, making it challenging to obtain real rewards.
>
> **An example scenario.** When **deploying a fine-tuned model for end-user applications**, users often provide natural language descriptions of their needs but may not include test cases (which can also be challenging for beginners or casual users). This situation is quite common and **real rewards are unobtainable**.
>
> **$\tilde{r}$ in such scenarios.** Our reward model $\tilde{r}$ doesn't require test cases from end users, and hence is applicable. Moreover, in many applications, such as GitHub Copilot, the model must offer a single best response, not a range of options. Thus, the ability to **rank programs without test cases (true rewards)** is not only realistic but also advantageous.
>
> **Additional benefits.** As discussed in Section 4, $\tilde{r}$ requires no additional training and adds minimal computational overhead, as $Q$ and $V$ are already evaluated during program generation. In contrast, the critic sampling strategy used in CodeRL and RLTF requires (i) training a separate repair model and (2) available test cases (i.e. true rewards) for post-processing, which is less desired due to aforementioned reasons.
>
> **Remark on training with $\tilde{r}$.** Recall that our reward model $\tilde{r}$ is obtained from $Q$ using Eq (19), without additional training. Using $\tilde{r}$ to train $Q$ from outset is in fact creating a chicken-and-egg problem. Besides, complex reward engineering during training is also something we tried to avoid.
>
>
> We hope our explanation addresses the reviewer's major concern and clarifies our choice.
>
> > The authors filter programs at evaluation time using a different reward than the one used during training. This makes it hard to interpret their performance numbers. Could it be the filtering that is mostly responsible for them?
>
> We hope our response to the main question helps address the concern about the use of a different reward. Ranking indeed improves performance as we expected. However, both initialization and conservative updates contribute significantly to the final performance. The comparison with CodeRL and RLTF with post-processing, i.e. critic sampling, is also available in Table 2. And we would like to highlight that our reward ranking is a more realistic post-processing strategy as explained above, and our model was trained with minimal reward designs.

---

> > ### Author Response · Authors · 2023-11-21
> > **Response to reviewer 3d3T [Part 2/3]**
> >
> > > The authors' method needs to be applied to a pre-trained LLM, which is used not only for initialisation but also for training. This is only possible with LLMs whose checkpoints are publicly available. In the case of the largest LLMs available today, even with a checkpoint available, considerable computational resources would be needed to apply the method.
> >
> > We understand the reviewer's concern, but we believe this computation concern also applies to other RL based fine-tuning methods. Although ours adds an additional $\phi$-stage compared to policy-based baselines, training $V_\phi$ is much easier than $Q$ or $\pi$, as $V$ is a function of $s$ while $Q$ and $\pi$ are function of $(s, a)$.
> >
> > > - The paper has a bit of an idiosyncratic structure ...
> > > - Some more proof-reading is needed as typos can be found here and there.
> >
> > We appreciate for the reviewer's constructive comments on our paper's presentation. We are delighted to implement these suggestions in the next revision, including but not limited to (1) adding an outline paragraph following the introduction for enhanced structure and clarity; (2) restructuring the related work section to ensure balanced and proportional content; and (3) conducting thorough proofreading to eliminate typographical and grammatical errors.
> >
> > ---
> > ### Other Questions
> >
> >
> > 1. We believe the reward engineering is a major factor, though the trade-off made by our conservative Bellman operator might also be one of the reasons. Polyak update of $q$ with small/conservative update rate can be a good practice we believe. The key insights we seek to convey are: (1) the reasons why Bellman optimality operator is not particularly practical, and (2) the rationale behind how conservative backups address this issue, which we believe are novel and non-trivial to the program synthesis task. However, it's always possible to further refine the trade-off between conservatism and performance, along with other engineering aspects.
> >
> >
> > 2. (**TL;DR**. The conservative operator avoids bootstrapping with overly optimistic value estimation. In addition, bounded value function and low initial TD-error (due to initialization) might also be helpful.)
> >
> >     For the possible divergence caused by *Deadly Triad*, many techniques used in current deep RL era aims to reduce the **overestimation bias**[1]. For example, Double DQN *``reduce overestimations by decomposing the max operation in the target into action selection and action evaluation''* [2]; TD3 [3] proposes clipped double Q-Learning along with some other modifications. When training with a pre-colleted dataset, especially with partial coverage, recent offline RL works **avoid bootstrapping with OOD actions**, (OOD denotes out-of-distribution), as those unseen actions could have arbitrarily large estimations, leading to diverging behavior. For example, TD3+BC [4] simply adds an regularization term penalizing deviation from the dataset.
> >
> >     Our methods resembles these above ideas: (1) we bound the value function by $R_{max}$, see Appendix D; (2) our conservative operator only select those actions with high probabilities $a'$ to evaluate target $Q(s', a')$, avoiding bootstrapping with OOD actions; (3) Our initialization may also alleviate this problem because it ensures value function starts with low bootstrapping error.
> >
> >
> > 3. Yes, it is indeed possible for a generic LLM. Adding an additional head can simply done by adding a linear layer subsequent to the decoder, akin to how the LM head is added. As the reviewer was asking $V^r(s)$, we assume the reviewer meant to ask how to add a residual head initialized to output zeros. To achieves this, one can add two heads $h_1$ and $h_2$ and let $h_1$ be trainable, $h_2$ be fixed, setting $V^r = h_1 - h_2$ achieves this functionality. We hope that we have correctly understood the question and  our answer clarifies it.
> >
> > 4. Those are the actual training curves. We hypothesize that it is because the initial checkpoint has been already fine-tuned on the training set, hence having a good training performance. Nonetheless, it highlights the difficulty of value-based learning and necessity of our proposed components.
> >
> > 5. Certainly, we are delighted to add an outline of paper organization at the end of our introduction section in revision, to improve our paper's clarity.
> >
> > 6. We are grateful for the suggestion and will accordingly update the caption to explicitly define $\ell$ and $p$.

---

> > > ### Author Response · Authors · 2023-11-21
> > > **Response to reviewer 3d3T [Part 3/3]**
> > >
> > > 7. Nucleus sampling was not used during training. We agree with the reviewer that it would be more suitable in Section 5 and will move it accordingly.
> > >
> > > 8. Certainly, we will explicitly define the abbreviations in the table captions.
> > >
> > > 9. The main reason that pass@1000 is missing in Table 2 is that they were not reported by the baselines, hence we cannot directly use the results. Evaluate pass@1000 for baselines are in fact very resources-intensive, especially considering that APPS has 5000 testing problems. As our resources were limited, we opted not to include the pass@1000 metric.
> > >
> > > 10. We hope our response to the primary concern addresses the reviewer's doubts regarding our use of the reward model $\tilde{r}$. We believe that our reward model provides practical benefits, given the aforementioned explanation. Nevertheless, we are happy to move part of the content to the appendix, saving some space to accommodate the reviewer's suggestions on presentation.
> > >
> > >
> > >
> > > [1] Van Hasselt, Hado, et al. "Deep reinforcement learning and the deadly triad." arXiv preprint arXiv:1812.02648 (2018).
> > >
> > > [2] Van Hasselt, Hado, Arthur Guez, and David Silver. "Deep reinforcement learning with double q-learning." Proceedings of the AAAI conference on artificial intelligence. Vol. 30. No. 1. 2016.
> > >
> > > [3] Fujimoto, Scott, Herke Hoof, and David Meger. "Addressing function approximation error in actor-critic methods." International conference on machine learning. PMLR, 2018.
> > >
> > > [4] Fujimoto, Scott, and Shixiang Shane Gu. "A minimalist approach to offline reinforcement learning." Advances in neural information processing systems 34 (2021): 20132-20145.

---

> ### Comment · Reviewer_3d3T · 2023-11-22
>
> I thank the authors for their exhaustive answers to my questions, as well as their receptiveness to my other suggestions for improving the paper.
>
> I can fully see their point about the unavailability of test case-based rewards in a realistic deployment setting. Still, I would suggest that an evaluation of their method with filtering based on environmental rewards $r(s,a)$ (or anyway the same reward function used for training it and carry out policy improvement) be added at least to the appendix. It might not be representative of a realistic deployment situation, but in my opinion it would be an insightful sanity check, also for any future readers.
>
> Also, have the authors already posted the revised version of their paper? I can still only see the initial version. I would like to read the revised one before I make any decisions on whether to update my score.

---

> ### Author Response · Authors · 2023-11-23
>
> Thank you for your prompt response and further suggestions.
>
> > Still, I would suggest that an evaluation of their method with filtering based on environmental rewards $r(s, a)$ be added at least to the appendix.
>
> Certainly, we will carry out the suggested experiments and include them in the Appendix in future revision.
>
> > Also, have the authors already posted the revised version of their paper? I can still only see the initial version.
>
> We are delighted to upload an initial revision as per the reviewer's request. In this preliminary revision, we have made the majority of the changes suggested by all the reviewers. The changes have been highlighted in blue. Specifically, we have made the following adjustments in line with Reviewer 3d3T's recommendations:
>
> 1. ``The paper has a bit of an idiosyncratic structure, with e.g. a whole section dedicated only to the reward filtering at evaluation time, a very small Related Work section, and no outline of the paper's contents in the introduction, which makes it a bit hard to read.''
>
>     - The introduction and related work sections have been revised for a more balanced structure.
>
>     - The reward section has been restructured as a subsection under our methodology section. To optimize space within this subsection, certain background paragraphs have been relocated to Appendix C, while other, less crucial details have been moved to Appendix E.
>
>
> 2. ``the authors do not elaborate on it. The authors should either: Justify this choice. ...''
>
>     - In Section 4.5, we added a paragraph to further elaborate on the deployment setting and to justify our choice.
>
>
> 3. ``Can the authors outline their paper's content and structure at the end of the intro?''
>
>     - We have added an outline of our paper's structure to the end of our introduction section.
>
> 4. ``In figure 3 (now figure 2 after revision), it would be helpful to explicitly define what and $\ell$ and $p$ stand for''
>
>     - The captions have been updated for explicit definitions.
>
> 5. ``Why is the Nucleus Sampling outlined at the end of section 3? In my opinion it would make more sense to have in section 5''
>
>     - The sampling details have been relocated to Section 5.
>
> 6. ``It would be helpful if the authors defined (e.g. in a caption) what "Intro", "Inter", "Comp", and "All", mean in tables 1 and 2.''
>
>     - The caption of Table 1 has been updated. While Table 2 remains unchanged at present, we plan to enhance our writing for conciseness, creating space for an updated caption for Table 2.
>
> 7. ``The study on generalisation of the reward model (notwithstanding my concerns on its use, detailed above) would better be moved to an appendix.''
>
>     - We have reduced the length of the 'reward model' subsection and moved some less important content to Appendix, to accommodate the reviewer's suggestions, which allowed us to maintain the 'reward model generalization' experiments in their current place. We hope that the reviewer finds this arrangement sensible after our clarification on choice of $\tilde{r}$. However, if the reviewer still believes these experiments would be better suited to the appendix, we are happy to make those revisions.
>
>
> Thank you once again for your thorough review and valuable insights! We hope that this initial revision adequately addresses the reviewer's concerns regarding the paper's overall structure and flow, even though it may still contain typographical and grammatical errors. We are committed to continuously refining our paper to improve its presentation.

---

> > ### Comment · Reviewer_3d3T · 2023-11-23
> >
> > I thank the authors for taking the time to post a preliminary revision of their paper; It is much more readable and better structured now.
> > All my concerns have been addressed and the paper has been revised as I suggested. I shall therefore raise my score to an 8.

---

### Official Review · Reviewer_F9ov · 2023-10-31

**Soundness:** 4 excellent
**Presentation:** 4 excellent
**Contribution:** 4 excellent
**Rating:** 8
**Confidence:** 2

**Summary:**

This paper derives a value-based approach for fine-tuning LLM's for code generation, motivated by the success of value-based methods in domains with similar constraints on data and reward modeling. The authors build on top of the Dueling DQN family of algorithms and design architectures and pretraining schemes for the Q/A/V functions that carefully incorporate the pretrained LLM policy in a way that provides mathematical guarantees that certain properties are met, such as ranking human solutions highly, leading to a policy function that is identical to the original LLM policy, and minimizing initial Bellman error. They then use a conservative Bellman operator based off greedy decoding, and show that they can derive a reward model from an inverse conservative Bellman operator without any additional training needed. Their approach is called Bellman-Coder. Experiments show that Bellman-Coder performs comparably to related work, and is often the best. In addition, the reward model generalizes to new domains and models.

**Strengths:**

- The paper is very well-written. The authors do an excellent job of discussing related work, motivating the design of a value-based approach for code generation, explaining the challenges behind designing a value-based approach, and describing how their design tackles these challenges. The technical sections are presented with care to make sure readers don't get lost in symbols, and I found them easy to understand (at least at a medium-to-high level) despite not being a RL expert. The experiments are presented well. Everything about the writing and clarity is excellent.
- The paper's approach is original and very high quality. It does appear that no prior work has leveraged value-based methods for code generation to the degree of success the authors present. Making progress on value-based methods for code generation seems like a significant research direction to pursue, given the importance of code generation as a primary application area of AI and the promise of value-based methods in other areas (as motivated by the authors). The only uncertainty of the significance might be due to the limited performance gains, but this is easily understandable given this paper seems to be the first to attempt it, and the uphill battle of designing such a method given the comparable ease of using a LLM as an off-the-shelf policy for policy-based methods. The authors took the harder route, and seem to have managed to get it to work comparably.
- I'd like to emphasize how high quality the mathematical derivations seem. The authors seem to have a very valuable expertise in RL and have put it to good use in making this method work. The ablations in figure 2 attest to the effectiveness of their formulation, with some caveats.

**Weaknesses:**

The primary weakness of the paper is that the results are not much better than prior work. However, given how difficult the perceived approach seems to be, I think this is okay.

I am not very up-to-date with the different techniques and evaluation metrics for RL code generation, so I can't comment too much on whether the evaluation is missing anything. But taking a quick look at related work, the authors seem to use the same benchmarks.

**Questions:**

In figure 2, why don't the approaches train for the same number of iterations? In the same vein, what do the dots signify on the dashed lines, and why do their distances change?

is it true that the reward model coming for free in section 4 is possible for any Bellman operator? what are the preconditions necessary for this? I ask because the "further comments" right before Proposition 3.1 made me think that the conservativeness of the Bellman operator defined was needed, but it sounds like in section 4 that it works for any Bellman operator. (I do not have much expertise in this area)

The related work paragraph in section 1 only discusses work on supervised LLM training and RL alternatives for sequence generation. given the amount of highly related work discussed in other sections of the introduction, it seems like this paragraph should be renamed to something more precise.

AlphaGo is described has having lots of off-policy data, but how does AlphaZero relate to the taxonomy of Figure 1? Isn't AlphaZero pretty much the same approach, but with all on-policy data?

---

> ### Author Response · Authors · 2023-11-21
>
> Thank you very much for your time in reviewing our work and for your positive assessment! We are glad that you found our work original and of very high quality, and our paper well-written! We are enthusiastic about improving our paper by incorporating your valuable comments.
>
> > Figure 2.
>
> We apologize for any ambiguity and inconsistency in the figure. The extended training of the other two veins was to demonstrate the challenges of value-based training, even when allowing more training episodes. Our evaluation points were randomly selected, leading to the varying distances between points. Despite these factors, we believed the overall trend was clear enough to demonstrate the impact of different components.
>
>
>
> > is it true that the reward model coming for free in section 4 is possible for any Bellman operator? what are the preconditions necessary for this?
>
> Yes, this is indeed possible for any operator that is a contraction. Being a contraction ensures a unique fixed point $Q$ upon convergence, which in turn allows an one-to-one correspondence with the reward function $\tilde{r}$. We will amend the comments preceding Proposition 3.1 to more clearly indicate that the property of being a contraction is a sufficient condition for Proposition 4.1.
>
>
> > The related work paragraph should be renamed to something more precise
>
> We certainly agree with the reviewer's point that the name *``related work''* is not quite fitting. We will rename it, for example, to *''RL in sequence generation''* or *''RL vs supervised fine-tuning''*.
>
> > how does AlphaZero relate to the taxonomy of Figure 1? Isn't AlphaZero pretty much the same approach, but with all on-policy data?
>
>
> Thank you for pointing out this aspect. The statement from the AlphaZero paper: *``Self-play games are always generated by using the latest parameters for this neural network''*, initially suggested to us the exclusive use of on-policy data. However, further examination led us to a pseudocode in their supplementary materials [1]. The code snippet suggests that they use a replay buffer, which is designed to re-use historical data in RL.
>
> For a training step $j$, data generated by $\pi_i$, where $i < j$, are also considered as **off-policy data** because they do not follow the state-action distribution $\rho_{\pi_j}(s, a)$ of the current policy $\pi_j$. Therefore, we believe that AlphaZero also uses a substantial amount of off-policy data, through experience replay.
>
>
>
> ```
> def train_network(config: AlphaZeroConfig, storage: SharedStorage,
>                   replay_buffer: ReplayBuffer):
>   network = Network()
>   optimizer = tf.train.MomentumOptimizer(config.learning_rate_schedule,
>                                          config.momentum)
>   for i in range(config.training_steps):
>     if i % config.checkpoint_interval == 0:
>       storage.save_network(i, network)
>     batch = replay_buffer.sample_batch()
>     update_weights(optimizer, network, batch, config.weight_decay)
>   storage.save_network(config.training_steps, network)
> ```
>
> [1] AlphaZero Supplementary Materials. https://www.science.org/doi/10.1126/science.aar6404#supplementary-materials

---

> > ### Comment · Reviewer_F9ov · 2023-11-22
> >
> > Thank you for your response and for addressing my questions!

---

### Official Review · Reviewer_R5vH · 2023-11-01

**Soundness:** 3 good
**Presentation:** 2 fair
**Contribution:** 3 good
**Rating:** 6
**Confidence:** 4

**Summary:**

this work proposes a value-function based training/inference procedure for program synthesis. rather than training a model (a policy) that ingest context (e.g. given as natural language comments) and generates tokens of code, this work leverages a value function Q(s,a) = V(s) + A(s,a), where V and A are implemented as neural networks.

In order to make Q learning work, this paper proposes two "tricks", one being a good Q-value initialization, the other being a conservative bellman update. Without these tricks, the training procedure would either not converge at all or improve very slowly.

Overall, this paper insists that they want to tackle the task of program generation by training a value function, and overcome a lot of challenges to make it work.

**Strengths:**

## quality: good
the overall quality of the paper is good, in that it even worked at all, and the lack of adhoc tricks.

### that it even worked at all

The biggest strength of this paper is that it has shown that code generation _can be_ done with a value network, despite the large number of state x action pairs needs to be considered, the unavailbility of a good initial Q function, and the volatility of the Q function "blowing up". Evaluation shown that the proposed method is on par with the policy based methods in terms of pass@k metrics.

The knowledge that value-based approach for code generation can be done _at all_ is a great merit, and cannot be under stated.

### lack of ad-hoc "tricks"

I find all the "tricks" presented in this work well founded and easy to understand. The reward hacking part is minimal and frankly very reasonable. Thus I am convinced this approach would work on different program synthesis domains just as well.

**Weaknesses:**

## significance

lack of evidence on why we need to do this to begin with

The following statement from the conclusion section is a very good motivation " it is recognized being sample-inefficient (see e.g. Nachum et al., 2017), meaning poorly using off-policy data, even data previously generated". Does this work actually demonstrate this with an experiment? I believe it should, to show (in a table) that the proposed method can "learn more with less", with x axis showing the amount of training data, and y-axis some performance metric. In case I have misread the paper and this is indeed one of the experiments, it should be top and foremost contribution this paper needs to claim. Currently this paper reads akin to "we did it because we can" but lacks the justification -- in the form of an experiment -- that this is something valuable to begin with.

I could think of many other benefits of having a value function, for instance, sometimes people might want a top-k ranked programs, which could be tricky to do based on only policy, but very intuitive with a value based model.

If the authors are able to justify their work with some experimental evidence, showing the advantages of a value-based modeling, this work will be that much better.

## clarity
this paper would benefit from a Figure1 that outline the overall "workflow" of the method. During reading of the paper, the phrase "which is yet another difficulty one must overcome to train a value function .... we introduce technique X to tackle this problem." The number of tricks quickly adds up, making the paper difficult to follow.

I feel the paper is overboard with definition and philosophy of "what is policy-based" and "what is value-based / RL". I believe these things have a very technical definition that the community can all agree with. The clean story could've been "off-policy learning through value iteration for generation would be good, as it use less data. here's how to do it, here's an experiment showing how well we can do as a function of amount of human data required".

Defining program synthesis as "generating program from natural language" is imprecise, as there are other works of "program synthesis" that generates program from input-output examples alone, or those that work with constraint solvers that generates programs from correctness specifications. A better wording could benefit here, as I instinctively reacted poorly by the opening definition of program synthesis, thinking it is way too narrow.

**Questions:**

## time to take sample?

The "go to" solution of program synthesis in the presence of given input-output examples (test cases) is to simply generate a vast number of programs, test all of them against spec through execution, and pick the ones that passed the test cases.

Thus, suppose I am a practitioner, the metric that I really care about is, given a synthesis problem, complete with natural language descriptions and a set of input-output test cases, and a time budget of 1 minute, can my synthesizer find a correct program?

This allows different methods to be compared head to head. A "dumb" enumerator -- for instance, a bigram generator -- might actually be reasonable of it can generate vast numbers of programs quickly, and a "smart" enumerator might actually be non-performative due to the time it takes to sample a full program to even check.

Thus, I would like to ask for the times it take to sample a full program on average, when compared against some of the other policy based methods.

---

> ### Author Response · Authors · 2023-11-21
> **Response to reviewer R5vH [Part 1/4]**
>
> Thank you very much for your time in reviewing our work and for your positive assessment! We are glad that you found our work of good quality, and showing the feasibility of value-based approach is a great merit. We are enthusiastic about improving our paper by incorporating your valuable comments.
>
> ---
>
> We would like to address the clarity and significance questions in a somewhat mixed order, as beginning with the technical definitions would help us better explain our motivations. We outline our responses as follows,
>
> 1. Our response to (a) clarifies the technical distinctions between (on-policy) policy-based and (off-policy) value-based methods, highlighting the data inefficiency of the former due to their inability to reuse historical data, a limitation rooted in their technical nature.
>
> 2. Our response to (b) lists empirical evidence from RL literature and public RL benchmarks.
>
> 3. Our response to (c) provides a demonstrative example to validate the intuition discussed in (a).
>
> 4. The rest are responses to the reviewer's other concerns.
>
> ---
>
> > (a) I feel the paper is overboard with definition and philosophy of "what is policy-based" and "what is value-based / RL". I believe these things have a very technical definition that the community can all agree with.
>
> As pointed out by the reviewer, we do realize that our description was quite verbal, which could create ambiguity and confusion. Our intended focus is on the algorithm's capability to **re-use previously generated data**, as being unable to reuse old/historical data (costly to collect) is very inefficient. Mathematically, this can be characterized by determining whether the objective function requires **data to be drawn from the distribution induced by the current policy**, as demonstrated in the following examples:
>
> | Algorithm | Category     | On/Off-Policy | Objective                                                                                                          |
> |-----------|--------------|---------------|-----------------------------------------------------------------------------------------------------------|
> | vanila PG | policy-based | on-policy     | $ L(\pi) = -\mathbb{E}_{\color{blue} \tau\sim\pi} [R(\tau)] $                                                      |
> | PPO       | policy-based | on-policy     | $ L(\pi) = -\mathbb{E}_{\color{blue} \tau \sim {\pi_k}} [ \frac{\pi(s, a)}{\pi_k(s, a)}\hat{A}(s, a) ] $    |
> | DQN       | value-based  | off-policy    | $ L(Q) = \mathbb{E}_{\color{blue} (s, a, s', r)\sim D} [(Q-\mathcal{B}^*Q)(s, a)]^2 $                              |
> | Ours      | value-based  | off-policy    |  $L(Q) = \mathbb{E}_{\color{blue} (s, a, s', r)\sim D} [(Q-\mathcal{B}^q Q)(s, a)]^2 $                             |
>
> where $R(\cdot)$ is cumulative trajectory-wise returns, $k$ denotes the current gradient/update step, $\tau \sim \pi$ or $\tau \sim \pi_k$ indicates the trajectories $\tau$ are rollouts collected by the **current policy** $\pi$ or $\pi_k$, whereas $D$ could be an **arbitrary dataset**.
>
> This technical difference was not precisely reflected in our current writing, as commented by the reviewer. We are gratefully for the reviewer's suggestion on technical definitions, which will largely help us to improve our paper's clarity.
>
> **Remarks on practical aspect.** (on-policy) policy gradient methods, in principle, cannot reuse previously generated, as they won't follow the current policy's (state-action) distribution. This is undesirable as generating samples with Transformers are known to be expensive.
>
> PPOCoder indeed has some discussion on this aspect. To briefly recap, they generate synthetic programs at $k=0$ (prior to fine-tuning), and subsequently fine-tune their model using both human and synthetic programs drawn from this mixed distribution. It was observed in PPOCoder (Figure 3(e)) that *an increase in synthetic programs negatively impacts fine-tuning performance*. This is partly because the mixed distribution doesn't align with the fine-tuned policy's state-action distribution.
>
> However, with more data, an ideal objective is generally expected to either improve or at least maintain a policy's performance, not decrease it. In general, the use of (growing) historical off-policy data is the key to the better sample efficiency observed in off-policy value-based methods.

---

> > ### Author Response · Authors · 2023-11-21
> > **Response to reviewer R5vH [Part 2/4]**
> >
> > > (b) The following statement from the conclusion section is a very good motivation " it is recognized being sample-inefficient (see e.g. Nachum et al., 2017), meaning poorly using off-policy data, even data previously generated". Does this work actually demonstrate this with an experiment?
> >
> > Although Nachum et al. [1] does not empirically demonstrate the sample inefficiency of policy-based methods, some empirical evidence could be found from other sources. For instance, [2] states that *``TD-style methods, such as off-policy actor-critic and Q-learning, are more sample-efficient ...''*, and supports this claim with empirical evaluation.
> >
> > Additional empirical comparisons can be found in public benchmarks, such as [OpenAI spinning up benchmark](https://spinningup.openai.com/en/latest/spinningup/bench.html), where policy-based methods often yield lower returns given the same number of environmental interactions. It can also be confirmed in some TD-learning works (e.g., [3, 4]), where policy-based baselines like PPO and TRPO often demonstrate lower sample efficiency.
> >
> >
> > > (c) I believe it should, to show (in a table) that the proposed method can "learn more with less", with x axis showing the amount of training data, and y-axis some performance metric.
> >
> > We believe the suggested experiment on comparison of number of data will be very insightful. This unfortunately creates two practical difficulties: it requires re-training our model and the baseline models for different synthetic data budgets, which would be overwhelming to us due to time and computational constraints during the discussion period. More importantly, a fair comparison is only achievable using an identical reward function design, as the reward is certainly a major factor of training efficiency, e.g. [5, 6].
> >
> >
> >
> > Nonetheless, we do believe that illustrating our motivation is important. To this end, we create a demonstrative experiment to illustrate empirical impact of re-using data with policy gradient objective. We hope the combination of
> >
> > - Response (a): an intuition given the technical definition
> > - Response (b): empirical evidence in RL community, (showing value-based ones *``learn more with less''*)
> > - Response (c): an empirical validation of (a)
> >
> > could addresses the reviewer's concern on our motivation.
> >
> >
> >
> > **TL;DR**. The experiment below shows that using  policy gradient objective with off-policy dataset may even decrease the performance of a policy. It implies that one should use on-policy data (and discard historical data) in principle, contributing to the undesired sample inefficiency. Hence, the experiment empirically validates our intuition-level explanation provided in response (a).

---

> ### Author Response · Authors · 2023-11-21
> **Response to reviewer R5vH [Part 3/4]**
>
> **A demonstrative experiment.**
>
> - **An illustrative MDP.** (An illustrative figure of this MDP can be found [anonymously here](https://www.dropbox.com/scl/fi/u9q95izjbphnfsngo2jit/mdp.pdf?rlkey=g11cvr12sp2kbsjudf8fju6ln&dl=0).)
>     Consider a chain MDP, where there are $n+1$ states $s_0, \ldots, s_n$ with two actions LEFT and RIGHT, where LEFT at $s_i$ proceed to the previous state $s_{i-1}$ with reward $r=1$, and RIGHT always proceed to the next state with  $r=-1$ except moving to the terminal state $s_n$, which has $r = k\times H$, where $k \geq 2$ and $H$ is the episode length. The optimal strategy should be always choosing RIGHT.
>
> - **Dataset.** We collect a dataset $D$ with an uniform policy with $\pi(\text{LEFT}|s) = \pi(\text{RIGHT}|s) = 0.5$ for all $s$. We will next use this dataset to simulate two scenarios: (1) dataset is on-policy: we optimize an uniform policy using this dataset, corresponding to optimizing an policy using on-policy dataset (as both are uniform); (2) we optimize a better policy $\pi_{new}(\text{RIGHT}|s)=0.75$ using this uniform dataset, corresponding to optimizing **an (new/improved) policy** using **old/historical dataset**.
>
> - **Training results.** (The training curves can be found [anonymously here](https://www.dropbox.com/scl/fi/usfy2awkn0tt2abkb0dih/training_curves.pdf?rlkey=64wkmw5lxdg4uv5b9yhs9vmss&dl=0).) (1) data is on-policy: Using $D$ to fine-tune an uniform policy, both PG and ours are able to improve the performance. (2) data is off-policy: We use $D$ to fine-tune a policy of $\pi_{new}(\text{RIGHT}|s)=0.75$. While the canonical PG loss is $ -\mathbb{E}_{\color{blue} \tau\sim\pi} [R(\tau)]$, reusing the old data (in program synthesis works) are often done by directly using the dataset's empirical distribution and the pseudo loss [7], i.e.
>
>     $L(\pi) = -\frac{1}{|D| H}\sum_{\color{blue} \tau \in D}\sum_{t=1}^H [\log \pi(a_t|s_t) R(\tau_{\geq t})]$.
>
>     PG only performs better than the dataset average performance but **lower than the average returns of $\pi_{new}$**, while ours could improves from $\pi_{new}$.
>
> - **Remark on this observation.** For a pair $(s, a)$, let's first define ''dataset average returns'', $\overline{R}(s, a) := \frac{1}{|D| H} \sum_{\tau\sim D} \sum_{t=1}^H [ \mathbb{I}(s_t=s) \mathbb{I}(a_t=a) R(\tau_{\geq t})]$, where $\mathbb{I}$ is an indicator function. In the case (2) described above, for a state $s$, optimizing $L(\pi)$ is equivalent to finding $\arg\max_a \overline{R}(s, a)$. It in fact has nothing to do with the initial policy's performance $J(\pi_{new})$ but is only determined by the dataset $D$, i.e. finding the actions with higher ``dataset average returns''.
>
>     Figure ([anonymously available here](https://www.dropbox.com/scl/fi/es0bekguugaoinpxo7isn/policies.pdf?rlkey=3td8423cjw6jfm0yp8167r7ap&dl=0)) validates that PG consistently converges to the actions with higher average ``dataset average returns'' $\overline{R}(s, a)$, instead of the optimal action, even when initialized from a near-optimal policy $\pi_{new}$. This demonstrates that training a new/improved policy using old/historical data can lead to a decrease in policy performance (which also explains the observation found by PPOCoder).

---

> ### Author Response · Authors · 2023-11-21
> **Response to reviewer R5vH [Part 4/4]**
>
> > I could think of many other benefits of having a value function, for instance, sometimes people might want a top-k ranked programs, which could be tricky to do based on only policy, but very intuitive with a value based model.
>
> This is indeed one of our considerations. The details, of selecting top-$k$ programs by value functions, are given in Section 4. In addition, for instance, one may also derive beam search strategy that maximizes values rather than likelihood and we certainly agree with the reviewer that having a value function create further opportunities.
>
> > Defining program synthesis as "generating program from natural language" is imprecise
>
> Thank you for pointing it out. We realize that we were narrowly focused in terms of program synthesis definition, because we were somewhat limited to the specific APPS dataset. It is very true that the program synthesis problem should be more general. We are delighted to revise our definition to be more accurate and encompassing, in line with the reviewer's suggestion.
>
>
> > Time to take sample?
>
> We take 5 problems from APPS test set (id 0 - 4) as examples and sampled 1000 programs for each, using one RTX 4090 24GB graphics card, with maximum batch sizes that do not create OOM errors. The table below shows the sampling time (in secs) per 1000 programs, in the format of mean$\pm$std. Our method takes approximately 1.7 to 2.0 times longer to sample.
>
> | Temperature | CodeRL (batch size = 66) | Ours (batch size = 44)|
> |---------|---------|---------|
> | 0.4 | 291.18$\pm$56.08 | 495.78$\pm$189.35 |
> | 1.0 | 290.88$\pm$56.20 | 595.84$\pm$65.66 |
>
>
> **Remark on faster sampling.** While we originally proposed using $Q$ as logits for sampling, it is also feasible to use the advantage $A$ as logits for sampling, especially when the goal is to generate as many outputs as possible without evaluating $Q$ and $V$ values. This would not require evaluating $V$, and therefore, we would expect similar sampling times. Under the same setting, sampling with advantage $A$ takes $275.79$ seconds and $322.78$ seconds for temperatures of $0.4$ and $1.0$, respectively, which are similar to CodeRL as we expected.
>
>
> ---
>
> [1] Nachum, Ofir, et al. "Bridging the gap between value and policy based reinforcement learning." Advances in neural information processing systems 30 (2017).
>
> [2] Gu, Shixiang, et al. "Q-Prop: Sample-Efficient Policy Gradient with An Off-Policy Critic." International Conference on Learning Representations. 2016
>
> [3] Haarnoja, Tuomas, et al. "Soft actor-critic: Off-policy maximum entropy deep reinforcement learning with a stochastic actor." International conference on machine learning. PMLR, 2018.
>
> [4] Fujimoto, Scott, Herke Hoof, and David Meger. "Addressing function approximation error in actor-critic methods." International conference on machine learning. PMLR, 2018.
>
> [5] Burda, Yuri, et al. "Exploration by random network distillation." Seventh International Conference on Learning Representations. 2019.
>
> [6] Harutyunyan, Anna, et al. "Hindsight credit assignment." Advances in neural information processing systems 32 (2019).
>
> [7] https://rail.eecs.berkeley.edu/deeprlcourse/deeprlcourse/static/slides/lec-5.pdf

---

> > ### Comment · Reviewer_R5vH · 2023-11-23
> > **keeping the score**
> >
> > thanks for the response! the work remains favorable after the response, and is a stronger form than what it was, especially the illustrative experiment.
> >
> > had there been more time we would loved to see the full experiment demonstrating the effectiveness of value-based learning, both on the learning side (comapred to policy based) and on the application side (perhaps some experiments on ranking, etc)

---

### Official Review · Reviewer_dB3F · 2023-11-03

**Soundness:** 3 good
**Presentation:** 3 good
**Contribution:** 4 excellent
**Rating:** 8
**Confidence:** 3

**Summary:**

The task is to generate code (sequence generation) given a natural language description, and to have that code pass the associated unit tests.

This is framed as an RL problem, and unlike most prior work it does not use an algorithm from the policy gradient family, but uses a sort of Q learning approach.

The main focus is on how to overcome the very long and large action space you get by treating each token as an action, and the sparse reward at the end. The main idea is to make good use of pre-trained language models that have been fine-tuned to generate code for given ground truth (description, code solution) pairs.

Using this is non-trivial however, and the paper takes a creative an impressive approach.

The main ideas are known but very nicely combined. Some highlights:
- the pretrained model is used to initialise a sensible Q function before the RL training
- decomposing Q(s,a) as state value + state-action advantage (dueling dqn)
- conservative bellman updates to stabilise the rl training loop.

A drawback of the approach is that it takes multiple steps, uses various tricky pre-training and residual fitting steps, and could be difficult to apply in practice. But this is not suprising since using Q learning here is challenging, and it is impressive the authors managed to make it work so well.

Detailed comments:

2.1 Policy: should say "assigns an action *from the set* $\Delta(\mathcal A)$" and generally this notation here needs checking.

In figure 2, why is zero training iterations the best? anyway this is a nice ablation / teaser for the paper.

Fig 3 is very cool indeed.

General question - since you sample a softened Q as your policy, you could actually apply policy gradients as well. What are the connections here? Could you combine your scheme with a policy gradient type of update?

**Strengths:**

See the summary.

**Weaknesses:**

See the summary.

**Questions:**

See the summary.

---

> ### Author Response · Authors · 2023-11-21
>
> Thank you very much for your time in reviewing our work and for your positive assessment! We are glad that you found our approach both creative and impressive! We are enthusiastic about improving our paper by incorporating your valuable comments.
>
> > 2.1 Policy: should say "assigns an action from the set $\Delta(\mathcal{A})$" and generally this notation here needs checking.
>
>
> Thank you for bringing this to our attention. We noticed that our notation was indeed inconsistent. In the statement *``assigns an action distribution $\Delta(\mathcal{A})$ ..., meaning predicting a token $\hat{w}_t$''*, the first part refers to a distribution over actions, while the latter suggests a single token. We'll revise our notation accordingly to  ensure better clarity and consistency.
>
> > In figure 2, why is zero training iterations the best? anyway this is a nice ablation / teaser for the paper.
>
> We hypothesize that it is because the initial checkpoint has been already fine-tuned on the training set, hence having a good performance. Nonetheless, as the reviewer pointed out, we believe it highlights the difficulty of value-based training and the necessity of the proposed components.
>
>
>
> > General question: since you sample a softened $Q$ as your policy, you could actually apply policy gradients as well. What are the connections here? Could you combine your scheme with a policy gradient type of update?
>
>
>
> As our focus was on value-based learning, we did not incorporate a policy gradient loss in our framework. However, we believe this could be a promising and exciting direction, as one may benefit from both the robustness of policy optimization and sample-efficiency of value-based learning, with careful designs.
>
> **Can PG loss be combined?** Yes, as suggested by the reviewer, it is certainly feasible to optimize $Q$-function with policy gradient objective through an energy-based policy, by computing $\nabla_\theta \log \pi_Q(a|s) = \nabla_\theta [Q_\theta(s, a)/\alpha - \log \sum_u \exp (Q_\theta(s, u)/\alpha)] $ (which is tractable due to a discrete vocabulary) and then plugging it into policy gradient.
>
>
> **Connection.** This question leads us to an earlier work [1], which exactly explores the combination of policy-based and value-based learning, by leveraging energy-based policies. This could also be considered as an actor-critic framework with a shared network ($s \to
>  \text{shared net} \to Q \xrightarrow{\mathrm{softmax}} \pi$ ), which is feasible when $\mathcal{A}$ is discrete. And the shared network can be optimized by the combination of policy gradient and temporal difference losses, applied to $\pi$ and $Q$ respectively.
>
>  We are thankful to the reviewer for this inspirational question, which creates an exciting opportunity. And we will include further discussion on this combination, along with relevant works, into our revision.
>
> [1] O'Donoghue, Brendan, et al. "Combining policy gradient and Q-learning." International Conference on Learning Representations. 2016."

---

### Meta-Review · Area_Chair_THDt · 2023-12-07

**Metareview:**

In this paper, the authors introduced a new RL-based approach for code generation (i.e. program synthesis). The new approach, B-Coder/ Bellman Coder, is a value-based approach utilizing pretrained LMs, a conservative Bellman operator, and learned value functions for post-processing generated programs. The experimental results demonstrate the potential of this approach on the challenging APPS benchmark.

All reviewers appreciated the right motivation and contribution of this work in successfully creating a challenging yet elegant value-based RL method for program synthesis tasks. The approach includes a nice combination of Q function and conservative Bellman updates. There are some minor concerns from the reviewers that the authors should try to address in their final revision, including the clarity of the technical details (e.g. policy-based vs. Value-based methods) and other presentation issues in the paper.

**Justification For Why Not Higher Score:**

The contribution of this work is quite significant as it offers a novel value-based RL method for program synthesis. Yet, the experimental results could be improved to have more evaluation on more than one benchmarks. The foundational models could also be improved to have applications in more variants of LMs (e.g. on GPT models).

**Justification For Why Not Lower Score:**

The current paper contains significant technical contributions that not much prior work has done before, e.g. a complex yet elegant value-based RL method for challenging code generation tasks. The paper is presented in a clear manner that would significantly benefit the research community in following and extending this research domain.

---

### Decision · Program_Chairs · 2024-01-16

Accept (spotlight)